# HE-SNR: Uncovering Latent Logic via Entropy for Guiding Mid-Training on SWE-BENCH

**Yueyang Wang** [1] [2]   **Jiawei Fu** [2]   **Baolong Bi** [2] [3]   **Xili Wang** [1] [2]   **Xiaoqing Liu** [2]

## Abstract

SWE-BENCH has emerged as the premier benchmark for evaluating Large Language Models on complex software engineering tasks. While these capabilities are fundamentally acquired during the *mid-training* phase and subsequently elicited during Supervised Fine-Tuning (SFT), there remains a critical deficit in metrics capable of guiding mid-training effectively. Standard metrics such as Perplexity (PPL) are compromised by the "Long-Context Tax" and exhibit weak correlation with downstream SWE performance. In this paper, we bridge this gap by first introducing a rigorous data filtering strategy. Crucially, we propose the **Entropy Compression Hypothesis**, redefining intelligence not by scalar Top-1 compression, but by the capacity to structure uncertainty into Entropy-Compressed States of low orders ("reasonable hesitation"). Grounded in this fine-grained entropy analysis, we formulate a novel metric, **HE-SNR** (High-Entropy Signal-to-Noise Ratio). We validate our approach on models with up to 560B parameters across different context windows (32K/128K). This work provides both the theoretical foundation and practical tools for optimizing the latent potential of LLMs in complex engineering domains.

## 1. Introduction

SWE-BENCH (Jimenez et al., 2023) has established itself as the definitive benchmark for evaluating the software engineering capabilities of Large Language Models (LLMs). Unlike traditional evaluations on isolated code snippets,

SWE-BENCH tasks models with resolving complex issues in real-world repositories, requiring agentic capabilities like strict instruction following, precise tool invocation, and multi-turn environment interaction. Acquiring these proficiencies typically involves Supervised Fine-Tuning (SFT) to align the model as an instruction-following agent. Because directly applying Reinforcement Learning (RL) to base models remains highly challenging for complex multi-turn tasks, recent SOTA frameworks (Wei et al., 2026; Wang et al., 2025a; Yang et al., 2025) uniformly build upon instruction-tuned models. Thus, an SFT phase is widely considered a crucial prerequisite for any meaningful assessment of model performance on the SWE-BENCH benchmark.

Extensive literature (Zhou et al., 2023; Ouyang et al., 2022) suggests that LLMs acquire fundamental reasoning capabilities during pre-training, while post-training merely formats these capabilities, often incurring an "alignment tax". To bridge this gap, recent research advocates for an intermediate *mid-training* stage (Roziere et al., 2023; Touvron et al., 2023), which focuses on high-density domains like code and mathematics to amplify logical reasoning and extend context windows. However, evaluating mid-training progress currently relies on post-SFT scores. This inherently lagging indicator severely hinders iteration efficiency due to exorbitant computational costs. Furthermore, alignment stochasticity during SFT often obscures the base model's true latent potential. Therefore, a robust mid-training metric that can directly assess the latent software engineering potential of base models is imperative.

While prior work links lower test set loss (e.g., PPL (Kaplan et al., 2020), BPC (Huang et al., 2024)) to downstream performance on benchmarks like MMLU or HumanEval (Hendrycks et al., 2020; Chen et al., 2021), these metrics remain confined to static, non-agentic tasks solvable via simple few-shot prompting (Brown et al., 2020). Similarly, although recent studies like LongPPL (Fang et al., 2025) address position bias in instruct models on long-context benchmarks (e.g., RULER (Hsieh et al., 2024)), they primarily identify tokens critical for information retrieval. When applied to agentic tasks like SWE-BENCH, which require complex SFT alignment, these existing metrics fail to capture the SFT-invariant signals that reflect a model's true

[1] School of Mathematical Sciences, Peking University, Beijing, China [2] Meituan, Beijing, China [3] Institute of Computing Technology, Chinese Academy of Sciences, Beijing, China. Correspondence to: Yueyang Wang <wangyueyang@stu.pku.edu.cn>, Jiawei Fu <fujiawei06@meituan.com>.

*Proceedings of the $43^{rd}$ International Conference on Machine Learning*, Seoul, South Korea. PMLR 306, 2026. Copyright 2026 by the author(s).

agentic potential. Given the absence of metrics tailored for agentic capabilities, PPL serves as our primary baseline.

Beyond its inability to capture agentic potential, the reliability of PPL as a mid-training guide is further compromised by a phenomenon we term the "Long-Context Tax". When extending context windows via frequency scaling strategies such as linear RoPE scaling (Su et al., 2024; Roziere et al., 2023) or YaRN (Peng et al., 2023), the model encounters an immediate distributional shift in positional embeddings. This shift temporarily inflates predictive entropy and degrades PPL (Chen et al., 2023; Hsieh et al., 2024), obscuring the model's actual progress. This discrepancy becomes increasingly pronounced as model scale increases. Consequently, to effectively steer the mid-training process, we must formulate a precise metric that isolates true reasoning signals and is robust against the Long-Context Tax.

In this work, we propose HE-SNR (High-Entropy Signal-to-Noise Ratio), a novel metric grounded in fine-grained entropy analysis. HE-SNR quantifies high-entropy decision points during mid-training and mitigates the Long-Context Tax, providing a robust feedback signal for agentic SWE tasks. We validate our approach on models ranging from tens of billions to hundreds of billions of parameters across extended context windows (32K/128K).

Our contributions are as follows:

- **Data-Efficient Evaluation Protocol.** We introduce a novel token-granular filtering strategy that achieves high correlation with SWE-BENCH scores using only **500 trajectories** (totaling $\approx$12.5M tokens).

- **Theoretical Insight: "Shift to** $\ln 3$**" & Entropy Compression Theory.** We propose the Entropy Compression Hypothesis, identifying a distinct distributional shift of high-entropy tokens towards $\ln 3$ as a fundamental signature of superior reasoning. This redefines intelligence as "reasonable hesitation" and elucidates why standard metrics like PPL fail to capture latent potential.

- **Novel Metric: High-Entropy Signal-to-Noise Ratio (HE-SNR).** Guided by this theory, we formulate HE-SNR. Validated across models with up to 560B parameters and a $10\times$ scaling span, HE-SNR effectively circumvents the "Long-Context Tax" and demonstrates a strict linear relationship with downstream capabilities, serving as a robust compass for mid-training.

- **Empirical Insight into the SFT Alignment Tax.** We reveal that SFT, while improving global PPL, significantly degrades performance on critical high-entropy tokens. This uncovers an intrinsic source of the "Alignment Tax" in software engineering tasks, suggesting that SFT prioritizes superficial pattern matching at the expense of complex reasoning capabilities.

**Conflict of Interest Disclosure** The authors Y.W., J.F., B.B., X.W., and X.L. are employed by or interning at Meituan, which leads the development of the LongCat-Flash-Lite and LongCat-Flash models evaluated in this paper.

## 2. Preliminaries

Given an input token sequence $\mathbf{x} = (x_1, \ldots, x_T)$, the task is to model the conditional probability of the next token $x_t$ given the context $x_{<t} = (x_1, \ldots, x_{t-1})$. Let $\mathcal{V}$ denote the vocabulary. At each step $t$, the model outputs a probability distribution $p_\theta(\cdot|x_{<t})$ over $\mathcal{V}$.

**Top-$k$ Candidate Set.** We define the candidate set $C_k(x_t)$ as the set of the top-$k$ tokens with the highest probabilities predicted by the model at step $t$. Formally:

$$C_k(x_t) = \{x_t^{(1)}, x_t^{(2)}, \ldots, x_t^{(k)}\} \qquad (1)$$

where the tokens are sorted in descending order of their predicted probabilities, i.e., $p_\theta(x_t^{(1)}|x_{<t}) \geq p_\theta(x_t^{(2)}|x_{<t}) \geq \cdots \geq p_\theta(x_t^{(k)}|x_{<t})$.

**Perplexity (PPL).** The standard PPL for a sequence $\mathbf{x}$ of length $T$ is defined as the exponentiated average negative log-likelihood of the target tokens:

$$\text{PPL}(\mathbf{x}) = \exp\left(-\frac{1}{T}\sum_{t=1}^{T} \ln p_\theta(x_t \mid x_{<t})\right) \qquad (2)$$

Additionally, we report Bits Per Character (BPC), which normalizes the negative log-likelihood by the sequence's character length rather than token count. This metric effectively functions as a vocabulary-agnostic PPL, enabling fair comparisons between models with different tokenizers.

**Top-$k$ Hit.** We define a *Top-$k$ Hit* at step $t$ as the event where the actual ground truth token $x_t$ is contained within the top-$k$ candidate set $C_k(x_t)$. This is formally expressed using the indicator function $\mathbb{I}(\cdot)$:

$$\text{Hit}_k(t) = \mathbb{I}(x_t \in C_k(x_t)) = \begin{cases} 1 & \text{if } x_t \in \{x_t^{(1)}, \ldots, x_t^{(k)}\} \\ 0 & \text{otherwise} \end{cases}$$
$$(3)$$

Accordingly, the sequence-level **Top-$k$ Accuracy** is the average hit rate:

$$\text{Acc}_k(\mathbf{x}) = \frac{1}{T}\sum_{t=1}^{T} \mathbb{I}(x_t \in C_k(x_t)) \qquad (4)$$

**Top-$k$ Miss.** Conversely, a *Top-$k$ Miss* denotes instances where the ground truth $x_t \notin C_k(x_t)$. The set of missed tokens is defined as $\mathcal{M}_k = \{x_t \mid x_t \notin C_k(x_t)\}$. In §4.2,

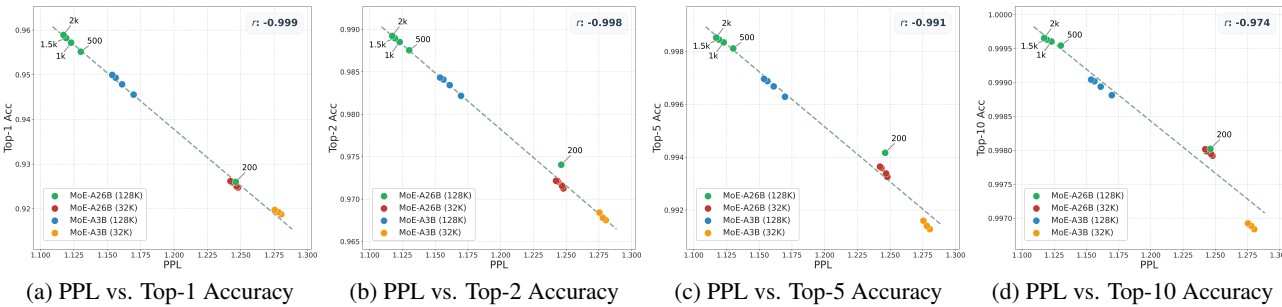

(a) PPL vs. Top-1 Accuracy      (b) PPL vs. Top-2 Accuracy      (c) PPL vs. Top-5 Accuracy      (d) PPL vs. Top-10 Accuracy

*Figure 1.* Correlation between PPL and Top-$k$ accuracy, evaluated on the LLM-generated components (*Thought* and *Action*) of the curated SWE-BENCH test dataset. Annotations (e.g., 500, 2k) indicate the training step count for specific checkpoints.

we refer to tokens in $\mathcal{M}_1$ and $\mathcal{M}_2$ as Non-Top-1 and Non-Top-2 tokens, respectively.

**Top-$k$ Entropy.** To measure the uncertainty strictly within the model's primary prediction space, we utilize the Top-$k$ Entropy:

$$H_{\text{topk}}(x_t) = -\sum_{i=1}^{k} \hat{p}_i(x_t) \ln \hat{p}_i(x_t) \qquad (5)$$

where

$$\hat{p}_i(x_t) = \frac{p_\theta(x_t^{(i)}|x_{<t})}{\sum_{j=1}^{k} p_\theta(x_t^{(j)}|x_{<t})}, \quad \forall i \in \{1, \ldots, k\}. \quad (6)$$

This metric quantifies the model's "hesitation" among its top candidates, ignoring long-tail probability mass. Given the rigorous demands of SWE tasks, we set $k = 10$ to strike a balance between computational efficiency and candidate coverage. Empirically, this setting is sufficiently inclusive, encompassing over 99.6% of target tokens in our test corpus (as shown in Figure 1d). Simultaneously, it avoids the prohibitive computational cost and noise interference associated with calculating entropy over the entire vocabulary.

## 3. Background and Problem Analysis

The prohibitive cost of mid-training and the delayed feedback from post-SFT evaluation create an urgent need for a reliable proxy metric to assess latent SWE potential. In this section, we analyze the failure of standard metrics like PPL, highlighting their inability to capture reasoning and their sensitivity to the "Long-Context Tax". Our empirical analysis leverages two distinct proprietary MoE models extended from 32K to 128K context: **MoE-A3B** (referred to as LongCat-Flash-Lite (Liu et al., 2026)), a base model with 68B total (3B active) parameters utilizing linear RoPE scaling; and **MoE-A26B** (referred to as LongCat-Flash (Meituan LongCat Team et al., 2025)), a large-scale model with 560B total (26B active) parameters employing YaRN.

### 3.1. Limitations of Perplexity

As shown in Figure 1, when evaluated on the LLM-generated components (*Thought* and *Action*) of the curated SWE-BENCH test dataset, PPL exhibits a strong linear correlation with Top-1 accuracy, a correlation that visibly diminishes as $k$ increases. Since Top-1 accuracy consistently exceeds 90%, PPL evidently measures replication precision rather than latent potential. Unlike SFT, which enforces a specific solution path, a base model only requires the target token to reside within a valid candidate set $C_k(x_t)$. Consequently, PPL misaligns with downstream SWE performance, as confirmed in Figure 2a.

This challenges the view that intelligence stems purely from the scalar compression of information, specifically, the minimization of loss (Huang et al., 2024; Delétang et al., 2023). Our findings suggest that raw compression often reflects rote repetition rather than reasoning. In §4.2.3, we elevate the compression hypothesis from a scalar information perspective to a distributional perspective involving entropy.

### 3.2. The Long-Context Tax

Extending the context window via base frequency scaling necessitates a readaptation phase to align the model with modified positional embeddings. However, this process incurs a transient performance regression we term the "Long-Context Tax". This phenomenon arises because altered rotational frequencies disrupt learned attention patterns, causing the attention mechanism to lose sharpness. Consequently, the inherent flattening of the softmax distribution manifests as a temporary spike in PPL.

Figure 2 tracks metric evolution during the 128K training phase on filtered *Action* tokens. While the Long-Context Tax is negligible for MoE-A3B, it is pronounced in the larger MoE-A26B: at Step 200, PPL and Top-1 accuracy degrade significantly, yet Top-10 accuracy remains stable alongside improved SWE-BENCH performance. This disparity suggests the tax erodes confidence in previously certain predictions rather than competence, shifting probability mass

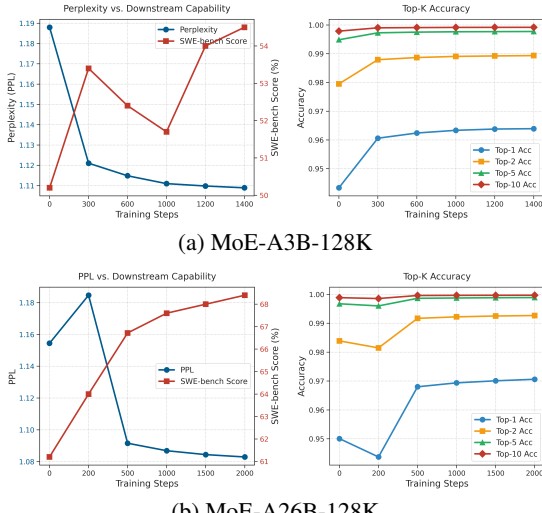

(a) MoE-A3B-128K

(b) MoE-A26B-128K

*Figure 2.* Evolution of metrics (on filtered Action tokens) vs. SWE-BENCH Pass@1 during 128K extension. "Step 0" marks the pre-RoPE adjustment 32K baseline. **(a)** MoE-A3B. **(b)** MoE-A26B. Note the inverse correlation in **(b)**: SWE performance improves despite PPL/Top-1 degradation caused by the Long-Context Tax.

without expelling the target from the candidate set $C_k(x_t)$. Thus, mid-training need not enforce Top-1 precision; ensuring the target resides within $C_k(x_t)$ suffices for retrieval during SFT. Consequently, raw PPL and Top-1 accuracy are misleading, underscoring the imperative for a robust metric that steers mid-training immune to the Long-Context Tax.

## 4. Methodology

In this section, we outline the formulation of our proposed mid-training metric. While sharing PPL's objective of quantifying the base model's comprehension of a test dataset, we diverge from its rigid assumption by rejecting the ground truth as an absolute standard in favor of treating it as a reference. By introducing strategic data selection and a granular token-level filtering mechanism, we formulate a metric from an entropy-based perspective that mitigates the "Long-Context Tax" while maintaining a high correlation with downstream SWE-BENCH performance.

### 4.1. Data Curation and Filtering Strategy

Since SWE capabilities are elicited during SFT, assessing them mid-training is challenging. To establish a predictive metric, we curate 500 successful trajectories from SWE-BENCH-VERIFIED, generated using the same synthesis methodology as the task-aligned SFT data to ensure strict distributional alignment and restricted to a 32K token limit for consistent evaluation with context windows of 32K or larger. Our curation adheres to a conservative filtering philosophy: prioritizing the exclusion of stylistic noise (e.g., SFT artifacts) to ensure the metric captures latent potential

rather than superficial imitation.

**Dataset Curation.** A multi-turn SWE trajectory can be formalized as $\tau = (o_1, r_1, a_1, \ldots, r_T, a_T)$. Given a codebase and issue description $I$, the agent leverages chain-of-thought reasoning ($r_t$) to ground its corrective actions ($a_t$), which are encapsulated in an XML-formatted protocol. At each turn, the environment returns an observation $o_{t+1}$ (e.g., execution logs). Detailed specifications are provided in Appendix B.

While the full trajectory serves as context, we compute metrics exclusively on *Action* components. Consistent with SFT objectives, we mask *Observations* (input context) and exclude *Thoughts*, which are often dominated by stylistic artifacts (e.g., filler words like "Now" or "Let"). This strategy prioritizes functional execution (what the model *does*) over narrative style (what it *says*), ensuring a significantly more rigorous assessment of latent utility.

**Action Filtering.** The *Action* component undergoes a rigorous filtering process to isolate functional tokens. First, we employ regular expressions to strip structural markers (e.g., XML tags) and markdown formatting, extracting the core content of commands, file paths, and code patches. For code segments, we utilize Abstract Syntax Tree (AST) parsing to specifically eliminate comments, as they represent natural language semantics distinct from executable logic. Finally, we apply a comprehensive noise reduction pass to remove redundant whitespace, newline characters, isolated symbols, and decorative formatting artifacts. To implement this, we annotate the raw text to identify these elements and subsequently map character-level tags to token granularity via offset alignment. This ensures the metric focuses purely on the functional correctness of the generated solution rather than formatting nuances.

### 4.2. Distribution of Top-10 Entropy

Figures 3 and 4 present the Top-10 entropy distributions of filtered action tokens across 500 trajectories for the MoE-A3B and MoE-A26B models at selected mid-training checkpoints. Additional results covering a broader range of checkpoints are provided in Appendix D for further reference.

The distributions are categorized into Top-1 hits, Top-1 misses, and Top-2 misses. Vertical red dashed lines mark reference entropy values ($\ln 2$ to $\ln 5$). For the Top-2 miss category, we highlight the global mode using a solid line, identified via Gaussian Kernel Density Estimation (KDE); comprehensive mathematical details regarding this process are provided in Appendix F.

We observe three distinct patterns in the entropy distributions:

- As expected, the entropy of Top-1 tokens is heavily concentrated near 0, reflecting the model's extremely high

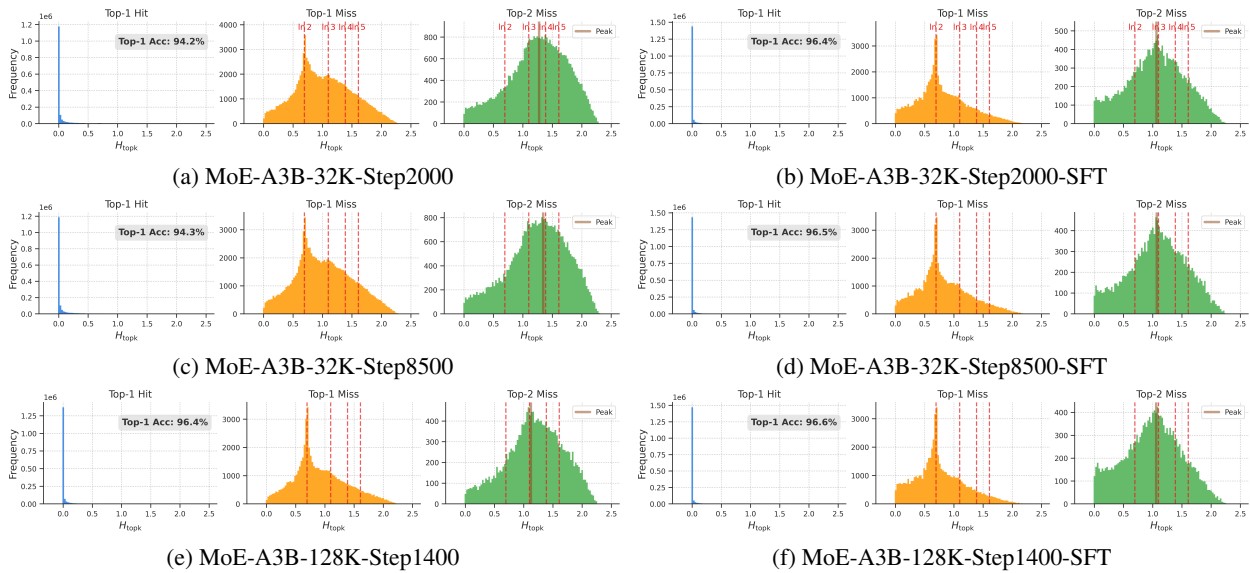

*Figure 3.* Top-10 entropy distributions for MoE-A3B: Base models (Left) vs. Post-SFT models (Right) across 32K/128K checkpoints. Red dashed lines mark $\ln 2$ to $\ln 5$. The brown vertical line in Non-Top-2 plots indicates the global peak (mode).

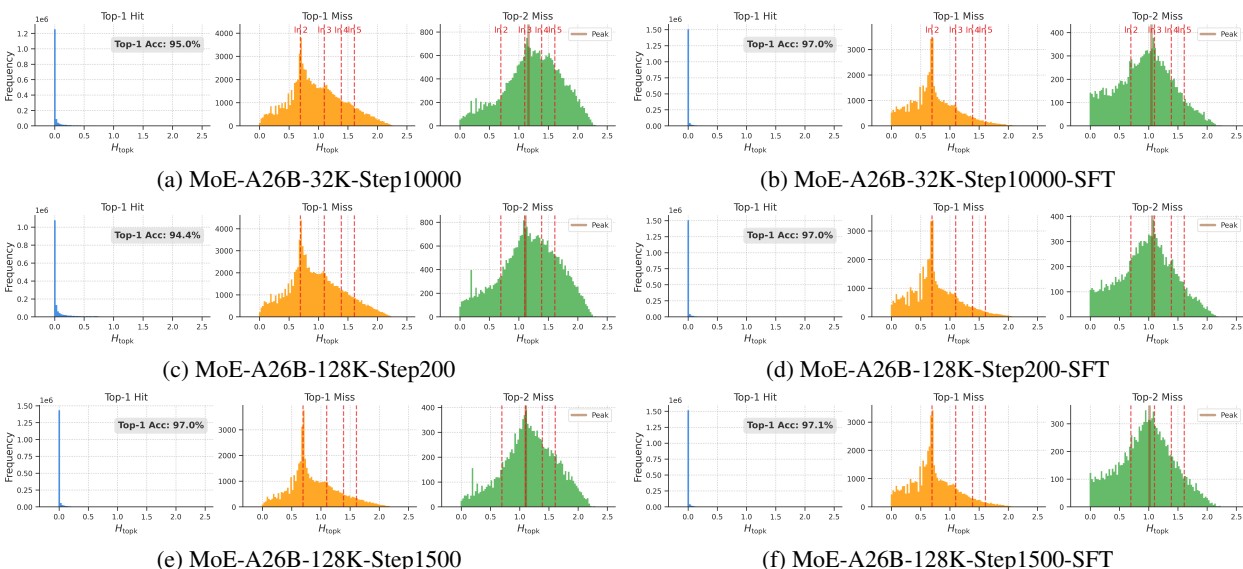

*Figure 4.* Top-10 entropy distributions for MoE-A26B: Base models (Left) vs. Post-SFT models (Right) at selected 32K and 128K checkpoints.

confidence in its correct predictions.

- For Non-Top-1 tokens, the distribution exhibits peaks at $\ln 2$ and $\ln 3$, with the $\ln 2$ peak being particularly prominent and consistent across scales.

- For Non-Top-2 tokens, weaker models exhibit a broad entropy distribution centered around $\ln 4$, whereas superior models display a distinct peak at $\ln 3$, a phenomenon we term the **"Shift to $\ln 3$"**. This shift indicates that superior models narrow their uncertainty to a smaller candidate set ($\sim 3$ options) even when missing the top-2 predictions, thereby effectively compressing the residual uncertainty.

### 4.2.1. THEORETICAL ENTROPY BOUNDS

**Lemma 4.1** (Maximum Entropy of Top-$k$ Distribution). *Let $X$ be a discrete random variable representing the model's next-token prediction distribution over the top-$k$ candidate set $C_k(x_t)$, with re-normalized probabilities $\hat{p}_i$ such that $\sum_{i=1}^{k} \hat{p}_i = 1$. The entropy $H(X)$ is bounded by:*

$$H(X) = -\sum_{i=1}^{k} \hat{p}_i \ln \hat{p}_i \leq \ln k \qquad (7)$$

*The equality holds if and only if the distribution is uniform, i.e., $\hat{p}_i = 1/k$ for all $i \in \{1, \ldots, k\}$.*

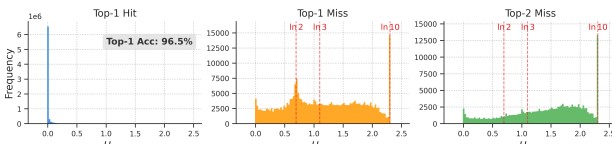

*Figure 5.* Top-10 entropy distributions for the MoE-A26B model computed on *Observation* tokens. The distribution exhibits distinct peaks near ln 2 and ln 10, contrasting with the structure observed in *Action* tokens.

The proof is straightforward and detailed in Appendix C. Despite its simplicity, this insight establishes a direct mapping between entropy values and the effective number of competing candidates $k$. It precisely elucidates the specific numerical peaks observed in our entropy distributions (e.g., $\ln 2 \approx 0.69$, $\ln 3 \approx 1.10$), providing a robust theoretical foundation for our subsequent analysis. Concrete case studies corresponding to these peaks are visualized and analyzed in Appendix G. Based on this mapping, we formalize the concept of the model's stable decision states:

**Definition 4.2** (Entropy-Compressed State). A prediction step $t$ is in this state of order $k$ if the probability mass is uniformly distributed over the top-$k$ candidates:

$$\sum_{i=1}^{k} \hat{p}_i \approx 1 \quad \text{and} \quad \hat{p}_i \approx 1/k, \quad \forall i \in \{1, \dots, k\}. \quad (8)$$

Empirically, these states manifest as distinct entropy peaks near $\ln k$. Under this framework, the conventional high-confidence state (Top-1 hit near $0 \approx \ln 1$) is the trivial collapse to order $k = 1$. We thus redefine model confidence: it is not solely indicated by low entropy ($\to 0$), but also by stable convergence to $\ln k$ for small $k$ (e.g., $k = 2, 3$). This state, where the model confidently vacillates among a refined set of options, is termed "reasonable hesitation."

#### 4.2.2. ENTROPY SHIFT TO ln 3 AND SFT EFFECTS

As illustrated in Figures 3 and 4, we observe a distinct distributional shift in entropy as the model evolves. During mid-training (especially context extension), the probability mass of Non-Top-2 tokens migrates from $\ln 4$ towards the Entropy-Compressed State at $\ln 3$. This indicates that as the model matures, it effectively compresses the candidate space, reducing ambiguity from a dispersed set of $k \geq 4$ options to a tighter cluster of 2–3 plausible candidates.

SFT further accelerates this compression, functioning as a potent entropy regularizer. Post-SFT distributions reveal a marked collapse of uncertainty, reinforcing the shift towards the $\ln 3$ boundary and below while significantly boosting Top-1 density. A striking example is the MoE-A26B model: while the base checkpoint at Step 200 (128K phase) exhibits a significant drop in Top-1 accuracy due to the Long-Context

Tax, this gap vanishes post-SFT, rendering their Top-1 levels nearly indistinguishable. However, the $\ln 3$ peak at Step 200 exhibits remarkable robustness to the Long-Context Tax.

This phenomenon confirms that Top-1 accuracy in the base model is a poor proxy for latent potential. While SFT successfully steers oscillating tokens into the Top-1 to Top-3 ranks, a result aligning with findings that SFT boosts confidence (Kadavath et al., 2022), we critically examine whether this confidence equates to correctness. In §5.5, we reveal that such aggressive regularization may actually impair intelligence in complex prediction scenarios, sacrificing reasoning capability for stylistic alignment.

**Generalizability of the Shift to** $\ln 3$. To ensure that the "Shift to $\ln 3$" is not merely an artifact of MoE architectures or specific to coding tasks, we extended our analysis to other model families and domains. Specifically, we evaluated Qwen2.5-72B-Base (Dense) and DeepSeek-V3-Base (MoE) on SWE-BENCH, as well as 5,000 rigorous Math QA samples across four base models. As detailed in Appendix E (Figures 13 and 14), all evaluated models consistently exhibit clear multi-modal entropy peaks at $\ln 1$, $\ln 2$, and $\ln 3$, along with the "Shift to $\ln 3$" trend. This confirms that the phenomenon reflects widespread reasoning behavior in structured logical tasks, rather than an architectural or dataset-specific artifact.

#### 4.2.3. ENTROPY COMPRESSION THEORY

Inspired by the observed distributional shift towards $\ln 3$, we propose the following theoretical framework:

**Hypothesis 4.3** (Entropy Compression Hypothesis). *Model optimization implicitly drives convergence to Entropy-Compressed States at natural boundaries $\ln k$. In rigorous logical domains, the higher the order $k$ of the Entropy-Compressed State a model preserves (following the structured progression $k = 1, 2, 3, \dots$), the more advanced its reasoning capability to resolve complex ambiguity.*

It is crucial to note that Entropy-Compressed States are not limited to low-order ranks ($k = 1, 2, 3$). As illustrated in Figure 5, the entropy distribution for MoE-A26B on *Observation* tokens lacks the characteristic $\ln 3$ peak, exhibiting instead a prominent peak at $\ln 10$. As detailed in Appendix G.3, this state stems from stochastic numerical sequences involving digits 0–9 (e.g., line numbers, random IDs). While this order-10 state reflects aleatoric uncertainty rather than error, it does not imply complex reasoning.

The absence of the $\ln 3$ peak, combined with the emergence of the $\ln 10$ peak, confirms that *Observation* sequences are dominated by stochasticity rather than structured logic, justifying their exclusion from metric computation. We thus posit that while the emergence of any Entropy-Compressed State signals a clear self-awareness of uncertainty, only

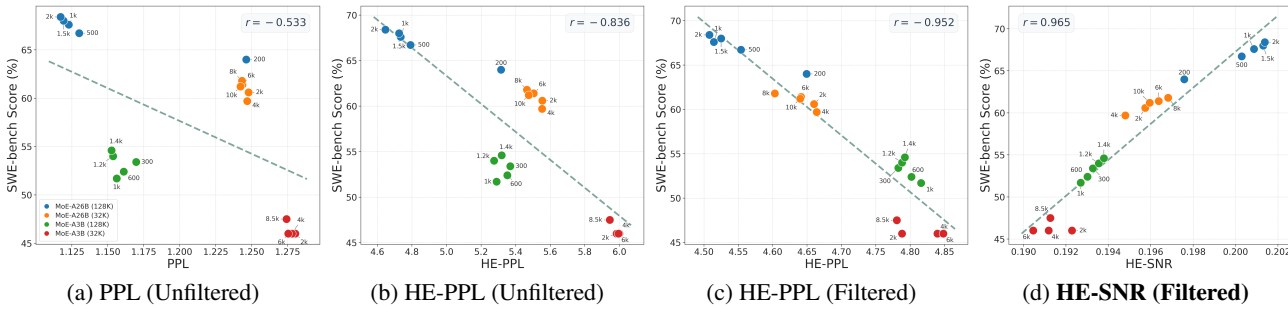

*Figure 6.* Correlation between mid-training metrics and downstream SWE-BENCH performance. We compare PPL, HE-PPL, and HE-SNR under **Unfiltered** (all tokens in Thought & Action) and **Filtered** (curated Action tokens only) settings. Notably, **HE-SNR (Filtered)** in **(d)** demonstrates superior linearity and robustness compared to PPL baselines. Annotations (e.g., 2k) denote training steps.

the capacity to sustain states of specific, progressively constrained orders (e.g., $k = 1 \rightarrow 2 \rightarrow 3\dots$) in logical contexts truly reflects advanced reasoning depth.

A comprehensive interpretation of our theory frames the training process as a continuum of entropy reduction. A randomly initialized model exhibits chaotic behavior with entropy approximating $\ln |\mathcal{V}|$, effectively an Entropy-Compressed State of the highest possible order $k = |\mathcal{V}|$. As training progresses and capability boundaries crystallize, predictive distributions converge towards lower natural boundaries $\ln k$, compressing local entropy away from chaos. Thus, the Entropy Compression Hypothesis can be fundamentally understood as the progressive compression of the valid candidate set size.

Hypothesis 4.3 refines the classic Compression-Intelligence Hypothesis. While compression is necessary for abstraction, focusing solely on scalar information minimization (as measured by PPL) merely quantifies memorization and repetition, evidenced by the strong linearity between PPL and Top-1 accuracy discussed in §3.1. True intelligence in complex reasoning involves retaining a structured distribution over valid reasoning paths. Our theory unifies these aspects, encapsulating both rote memorization ($k = 1$) and deliberative reasoning ($k > 1$) within the framework of Entropy-Compressed States.

### 4.3. Metric Formulation and Threshold Selection

Synthesizing the preceding analyses, we propose the **High-Entropy Signal-to-Noise Ratio (HE-SNR)**. Unlike PPL, which measures the absolute probability, HE-SNR evaluates the relative strength of the target signal against the background noise:

$$\text{HE-SNR} = \frac{1}{|\mathcal{H}|} \sum_{t \in \mathcal{H}} \frac{p(x_t)}{H_{\text{top10}}(x_t)}, \quad (9)$$

where $\mathcal{H}$ denotes the **High-Entropy Decision Set**:

$$\mathcal{H} = \{t \mid H_{\text{top10}}(x_t) > \epsilon \text{ and } x_t \in C_{10}(x_t)\}. \quad (10)$$

Here, $p(x_t)$ represents the signal, while $H_{\text{top10}}(x_t)$ represents the noise. The ratio quantifies how prominently the target stands out from the uncertainty floor. Crucially, the condition $x_t \in C_{10}(x_t)$ restricts evaluation to instances where the ground truth is plausible. This filters out tokens with severe distributional divergence, often caused by superficial stylistic mismatches rather than semantic errors, preventing outliers from skewing the metric.

**Threshold Selection.** Superior models naturally exhibit a distributional shift towards the $\ln 3$ peak, a tendency significantly intensified by SFT. Consequently, effective forecasting requires focusing on the recalcitrant uncertainty that resists SFT regularization. To operationalize this, we establish a critical entropy threshold $\epsilon$ at the boundary distinguishing 3-candidate uncertainty from that of 4 or more:

$$\epsilon = \frac{\ln 3 + \ln 4}{2} \quad (11)$$

This threshold delineates our High-Entropy Decision Set ($\mathcal{H}$), effectively resolving the indeterminacy where explicit Entropy-Compressed States have not yet crystallized. By focusing on this regime, we isolate critical reasoning points and capture persistent reasoning challenges that remain unresolved even after aggressive SFT regularization.

## 5. Main Results

### 5.1. Metric Comparison and Analysis

**Experimental Setup.** We evaluate multiple mid-training checkpoints from both MoE-A3B and MoE-A26B models across 32K and 128K context phases. For each checkpoint, we perform SFT using over 10,000 SWE-related trajectories for 3 epochs. To ensure robustness, the downstream performance on SWE-BENCH-VERIFIED is reported as the Pass@1 score, averaged over three independent evaluation runs to guarantee the statistical reliability of the results.

Analogous to the formulation of HE-SNR, we define **High-Entropy PPL (HE-PPL)** as the PPL computed exclusively over the high-entropy token set $\mathcal{H}$ (Equation 10). Figure 6

*Table 1.* Ablation Study on Token Filtering Strategies. We measure the impact of progressive filtering steps on the correlation between HE-SNR and SWE-BENCH scores. While Pearson $r$ measures the linear fit, Kendall $\tau$ captures ranking consistency. Our full pipeline achieves optimal performance across both metrics.

| Token Type | Filtering Strategy | Pearson $r$ | Kendall $\tau$ |
|---|---|---|---|
| Thinking | None (Raw Tokens) | 0.5581 | 0.5192 |
| Action | None (Raw Action Tokens) | 0.9666 | 0.9440 |
| Action | + Remove XML formatting tags | 0.9526 | 0.9558 |
| Action | + Remove redundant whitespace & symbols | 0.9520 | 0.9676 |
| Action | + AST-based comment removal (Full Pipeline) | 0.9649 | 0.9794 |

presents a comprehensive correlation analysis between these mid-training metrics and downstream SWE-BENCH performance, yielding several critical insights:

- **Failure of Standard PPL.** First, as shown in Figure 6a, standard PPL computed over all tokens (Thought and Action) exhibits poor correlation with downstream performance. Notably, it suffers severely from the "Long-Context Tax": the MoE-A26B-128K-Step200 checkpoint shows a PPL comparable to the 32K baseline, despite a significant improvement in actual SWE capabilities.

- **Efficacy of Data Filtering.** Second, comparing HE-PPL in the unfiltered setting (Figure 6b) versus the filtered setting (Figure 6c), the latter demonstrates a markedly stronger linear relationship. This validates our hypothesis that filtering out the thought part and stylistic noise effectively isolates the signal predictive of latent capability, thereby providing a more reliable indicator of downstream potential.

- **Superiority of HE-SNR.** Finally and most crucially, Figure 6d demonstrates that HE-SNR effectively mitigates the Long-Context Tax while maintaining strict rank consistency and linearity across the vast majority of checkpoints. This robustness establishes HE-SNR as a superior indicator to standard PPL, which fails to decouple transient entropy spikes from genuine capability growth.

## 5.2. Ablation Study on Token Filtering Strategies

To reliably predict post-SFT performance, our metric must isolate SFT-invariant signals by filtering stylistic artifacts. We conducted an ablation study to evaluate the impact of our data curation pipeline on the correlation between HE-SNR and SWE-BENCH downstream performance.

As shown in Table 1, shifting the evaluation from "Thinking" tokens to "Action" tokens drastically improves both correlation metrics, with Pearson $r$ increasing from 0.5581 to 0.9666 and Kendall's $\tau$ from 0.5192 to 0.9440. This is because the "Thinking" portion of the trajectory is heavily dictated by SFT prompt templates, making its entropy less predictive of ultimate task success than the strictly executable "Action" part. Furthermore, the progressive application of our filtering strategies, including removing XML

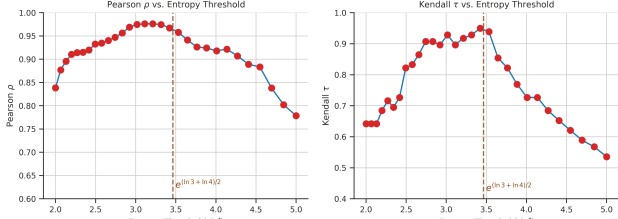

*Figure 7.* Impact of different entropy thresholds on the correlation between HE-SNR and downstream SWE-BENCH scores. We evaluate both Pearson (linear correlation) and Kendall (crucial ranking correlation) across a threshold range of $\ln t$, where $t \in [2, 5]$. As demonstrated, our empirically chosen threshold of $\epsilon = (\ln 3 + \ln 4)/2$ is remarkably close to the optimal threshold for both metrics, confirming the robustness of our metric design.

formatting tags, redundant whitespace, and AST-based comments, strictly isolates the executable logic. This full filtering pipeline pushes the crucial ranking consistency (Kendall $\tau$) to its peak of 0.9794, demonstrating that stripping away these SFT-induced artifacts is essential for constructing a robust mid-training metric.

## 5.3. Robustness of Threshold Selection

To validate the generalizability and robustness of our chosen entropy threshold $\epsilon = (\ln 3 + \ln 4)/2$, we conducted a sensitivity analysis examining the impact of different threshold values on the correlation between HE-SNR and downstream SWE-BENCH scores.

As illustrated in Figure 7, we evaluated both Pearson $r$ (linear fit) and Kendall $\tau$ (ranking consistency) across a continuous threshold range corresponding to $\ln t$, where $t \in [2, 5]$. The results demonstrate that our observation-based threshold is remarkably close to the empirical optimum for both metrics. More importantly, the correlation curves exhibit a robust plateau around this value, indicating that the metric is not overfitted to a single "magic number"—any threshold within this neighborhood yields highly competitive correlations. This confirms that our entropy-based threshold methodology is structurally sound and can be robustly extended to evaluate reasoning capabilities without requiring exhaustive hyperparameter tuning.

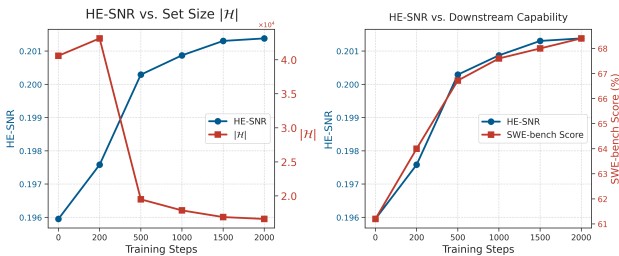

*Figure 8.* Evolution of $|\mathcal{H}|$ and HE-SNR for MoE-A26B during 128K extension. "Step 0" marks the pre-RoPE baseline. Note that while $|\mathcal{H}|$ spikes due to the Long-Context Tax, HE-SNR maintains a robust upward trend aligned with downstream capability growth.

### 5.4. Robustness of SNR Against the Long-Context Tax

Figure 8 tracks the size of the High-Entropy Decision Set $|\mathcal{H}|$ and HE-SNR for the MoE-A26B model during 128K training. We observe that $|\mathcal{H}|$ spikes at Step 200, confirming that the Long-Context Tax inflates the volume of uncertain tokens, which in turn degrades absolute metrics like HE-PPL. In contrast, HE-SNR maintains a consistent upward trend closely aligned with downstream performance. By measuring the relative signal-to-noise ratio, HE-SNR normalizes inherent global entropy shifts. This suggests that mid-training evaluation should prioritize the model's *degree of mastery* (SNR) over difficult tokens, rather than the mere *quantity* of such "hard problems" encountered.

### 5.5. The Entropy Cost of SFT: A Source of Alignment Tax

Our previous analysis established that prediction accuracy on tokens within the High-Entropy Decision Set is strongly correlated with downstream capabilities. While SFT successfully instills fixed patterns and reduces global PPL (Figure 9a), this alignment appears to come at a cost. Figures 9b and 9c reveal that within the set $\mathcal{H}$ (the subset most predictive of SWE success), both HE-PPL and HE-SNR degrade post-SFT. This phenomenon offers a potential explanation for the "Alignment Tax" in agentic tasks: to acquire specific stylistic patterns during SFT, the model may sacrifice prediction accuracy on critical high-entropy tokens, potentially impairing its reasoning capability in complex prediction scenarios. Our findings suggest that the Alignment Tax manifests prominently in the High-Entropy Decision set, offering a novel perspective for understanding and mitigating this trade-off in future alignment strategies.

## 6. Discussion and Future Work

**Discussion.** We present a framework for assessing latent SWE potential during mid-training, introducing a rigorous data filtering strategy and the HE-SNR metric, which robustly predicts post-SFT performance. By proposing the

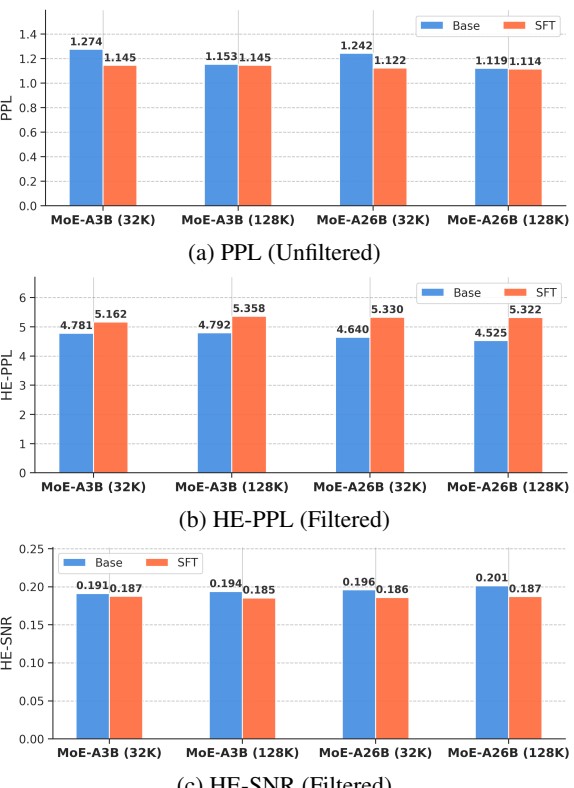

*Figure 9.* Impact of SFT on metric performance. **(a)** Global PPL improves post-SFT, reflecting pattern learning. Conversely, high-entropy metrics **(b)** HE-PPL and **(c)** HE-SNR degrade. This divergence suggests that SFT prioritizes rote pattern matching at the expense of reasoning on critical decision points.

Entropy Compression Hypothesis, we redefine intelligence as the compression of uncertainty into a refined candidate set. Our analysis reveals that latent potential is encoded in high-entropy tokens, elucidating the SFT "Alignment Tax" and offering a novel perspective for reasoning optimization.

**Future Work.** HE-SNR currently relies on a static threshold and data-specific patterns. Since SFT is sensitive to code style, future work will explore code canonicalization or style transfer to ensure high-entropy tokens reflect logical uncertainty rather than stylistic divergence. Furthermore, we aim to develop adaptive thresholding mechanisms to capture varying convergence rates. Crucially, we plan to extend validation to diverse architectures and broader logical domains (e.g., mathematics), specifically investigating the emergence of higher-order Entropy-Compressed States ($k \geq 4$) in tasks with greater intrinsic complexity.

## Impact Statement

This paper presents work aimed at advancing the evaluation and optimization of Large Language Models, specifically in the domain of software engineering. By introducing

a metric (HE-SNR) that accurately predicts downstream performance during mid-training, our work contributes to computational efficiency, potentially reducing the energy consumption and carbon footprint associated with training large-scale models by identifying sub-optimal candidates early. Furthermore, by emphasizing latent reasoning over rote memorization, our approach aims to foster the development of more robust and reliable coding assistants. We do not foresee immediate negative societal consequences or specific ethical issues beyond the general considerations associated with the automation of software development.

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

# A. Related Works

**Limitations of PPL and Truncation Strategies.** Perplexity (PPL) has long served as the de facto metric for evaluating language models. However, its reliability is increasingly questioned, particularly in long-context scenarios where it fails to accurately reflect effective context utilization (Hsieh et al., 2024) or distinguish between rote memorization and genuine reasoning. Recent studies, such as LongPPL (Fang et al., 2025), share the high-level insight that standard PPL overlooks key tokens in long-context scenarios. However, LongPPL primarily addresses position bias in instruct models for information retrieval tasks (e.g., RULER). In contrast, our work focuses on base models and extracts SFT-invariant signals to predict complex logical reasoning in agentic tasks. Given their fundamentally different evaluation targets (retrieval vs. reasoning), metrics like LongPPL may struggle to predict complex agentic performance. To mitigate the impact of the unreliable "long tail" in the probability distribution, generation strategies such as Top-$k$ (Fan et al., 2018) and Nucleus (Top-$p$) sampling (Holtzman et al., 2020) have become standard. These methods operate on the premise that the model's trustworthy signal is concentrated in the high-probability region, while the tail often contains noise or hallucinations. This shift from evaluating the full distribution to focusing on a truncated candidate set provides a foundational motivation for our work: moving beyond scalar PPL to metrics that respect the valid candidate boundary.

**Entropy as a Proxy for Reasoning.** Beyond simple probability truncation, entropy has gained prominence as a more nuanced proxy for quantifying uncertainty and reasoning branching in LLMs. Prior research demonstrates that LLMs are generally well-calibrated, exhibiting high entropy specifically at knowledge boundaries (Kadavath et al., 2022). Recent studies have further linked entropy patterns to reasoning efficacy. For instance, high-entropy tokens have been identified as critical "forking points" that steer reasoning pathways, with findings showing that optimizing these minority tokens alone drives significant performance gains (Wang et al., 2025b). Similarly, leveraging internal confidence signals to filter low-quality reasoning traces has proven effective in enhancing efficiency (Fu et al., 2025). These works suggest that intelligence is not uniformly distributed across all tokens but is concentrated in specific high-entropy decision points.

**The Alignment Tax.** While Supervised Fine-Tuning (SFT) is essential for eliciting capabilities and aligning models with human intent, it often incurs a performance regression known as the "Alignment Tax" (Ouyang et al., 2022; Askell et al., 2021). This phenomenon typically manifests as a reduction in generation diversity or a degradation in tasks requiring complex reasoning. Recent theoretical perspectives, such as the *Superficial Alignment Hypothesis* (Zhou et al., 2023), posit that SFT primarily adapts the model to a specific stylistic format rather than imparting new knowledge. Complementary lines of work mitigate this regression from the data side via expert-guided pre-training data refinement (Bi et al., 2025a), or from the post-training side via structured rubric-based reward signals that promote exploration (Bi et al., 2025b).

# B. SWE-bench Task Formulation and Trajectory Details

SWE-BENCH (Jimenez et al., 2023) evaluates LLMs on real-world software engineering tasks sourced from popular Python repositories like `django` and `scikit-learn`. Unlike traditional code generation benchmarks (e.g., HumanEval (Chen et al., 2021)) that focus on self-contained function synthesis, SWE-BENCH requires models to resolve genuine GitHub issues (e.g., bug reports) within a full repository context.

**Mathematical Formulation.** Formally, given a codebase and an issue description $I$, the model operates as an autonomous agent within a Dockerized environment to generate a solution patch. We model this process as a multi-turn trajectory:

$$\tau = (o_1, r_1, a_1, \ldots, r_T, a_T) \tag{12}$$

where the initial observation $o_1$ corresponds to the issue description $I$. In each interaction turn $t$, the agent leverages chain-of-thought reasoning ($r_t$) to ground its corrective actions ($a_t$). These actions (e.g., `search_code`, `edit_file`) are encapsulated in an XML-formatted protocol. The environment subsequently returns an execution observation $o_{t+1}$ (e.g., code content or error logs).

The final submission is verified against a hidden test suite, making the evaluation computationally intensive and heavily reliant on complex agentic capabilities. This pipeline demands precise code generation and long-context reasoning. Figure 10 illustrates the workflow of a single interaction turn within this trajectory.

# C. Proof of Maximum Entropy Bound (Lemma 4.1)

In this section, we provide the formal derivation for the upper bound of the Top-$k$ entropy as presented in Lemma 4.1.

**Observation:**
```
Execution output of [str_replace_editor]:
[File: testbed/django/template/base.py]

...

124: register = Library()
125:
126: class ExtendsNode():
127:     must_be_first = True

...
```

**Thought:**
Perfect! Now I understand the complete picture. The issue is simply that `ExtendsNode` doesn't inherit from `Node`, so it doesn't have the `render_annotated` method.
Now let me implement the fix.

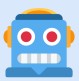

**Action:**
```
<function=str_replace_editor>
<parameter=command>str_replace</parameter>
<parameter=path>/testbed/django/template/loader_tags.py</parameter>
<parameter=old_str>class ExtendsNode():</parameter>
<parameter=new_str>class ExtendsNode(Node):</parameter>
</function>
```

*Figure 10.* Visual representation of a single interaction turn in the SWE-BENCH trajectory. The agent synthesizes the environmental observation ($o_t$) and chain-of-thought reasoning ($r_t$) to generate a grounded, XML-formatted action ($a_t$).

*Proof.* Let $X$ be a discrete random variable representing the model's next-token prediction distribution over the top-$k$ candidate set $C_k(x_t)$, with re-normalized probabilities $\hat{p}_i$ such that $\sum_{i=1}^{k} \hat{p}_i = 1$. The entropy is defined as:

$$H(X) = -\sum_{i=1}^{k} \hat{p}_i \ln \hat{p}_i = \sum_{i=1}^{k} \hat{p}_i \ln\left(\frac{1}{\hat{p}_i}\right) \tag{13}$$

Since the natural logarithm function $f(y) = \ln y$ is strictly concave, we can apply **Jensen's Inequality**, which states that for a random variable $Y$ and a concave function $f$, $\mathbb{E}[f(Y)] \leq f(\mathbb{E}[Y])$.

Let us define a random variable $Y$ that takes the value $1/\hat{p}_i$ with probability $\hat{p}_i$. Then, the entropy $H(X)$ can be viewed as the expectation $\mathbb{E}[\ln Y]$. Applying Jensen's Inequality yields:

$$\mathbb{E}\left[\ln\left(\frac{1}{\hat{p}_i}\right)\right] \leq \ln\left(\mathbb{E}\left[\frac{1}{\hat{p}_i}\right]\right) \tag{14}$$

Substituting the expectation formula:

$$\sum_{i=1}^{k} \hat{p}_i \ln\left(\frac{1}{\hat{p}_i}\right) \leq \ln\left(\sum_{i=1}^{k} \hat{p}_i \cdot \frac{1}{\hat{p}_i}\right) \tag{15}$$

Simplifying the term inside the logarithm:

$$\sum_{i=1}^{k} \hat{p}_i \cdot \frac{1}{\hat{p}_i} = \sum_{i=1}^{k} 1 = k \tag{16}$$

Thus, we obtain the bound:

$$H(X) \leq \ln k \tag{17}$$

The equality condition for Jensen's Inequality holds if and only if the random variable $Y$ is constant (almost surely). This implies that $1/\hat{p}_i = c$ for all $i$, which necessitates $\hat{p}_i = 1/k$. Therefore, the entropy is maximized when the distribution is uniform over the $k$ candidates. $\qquad\square$

## D. Additional Entropy Distribution Analysis

In this section, we present the Top-10 entropy distributions for a broader range of mid-training checkpoints. As illustrated in Figures 11 and 12, these extended visualizations provide clearer empirical evidence of the "Shift to $\ln 3$" phenomenon, particularly for **Non-Top-2 tokens**, where the probability mass progressively concentrates around $\ln 3$ as training advances.

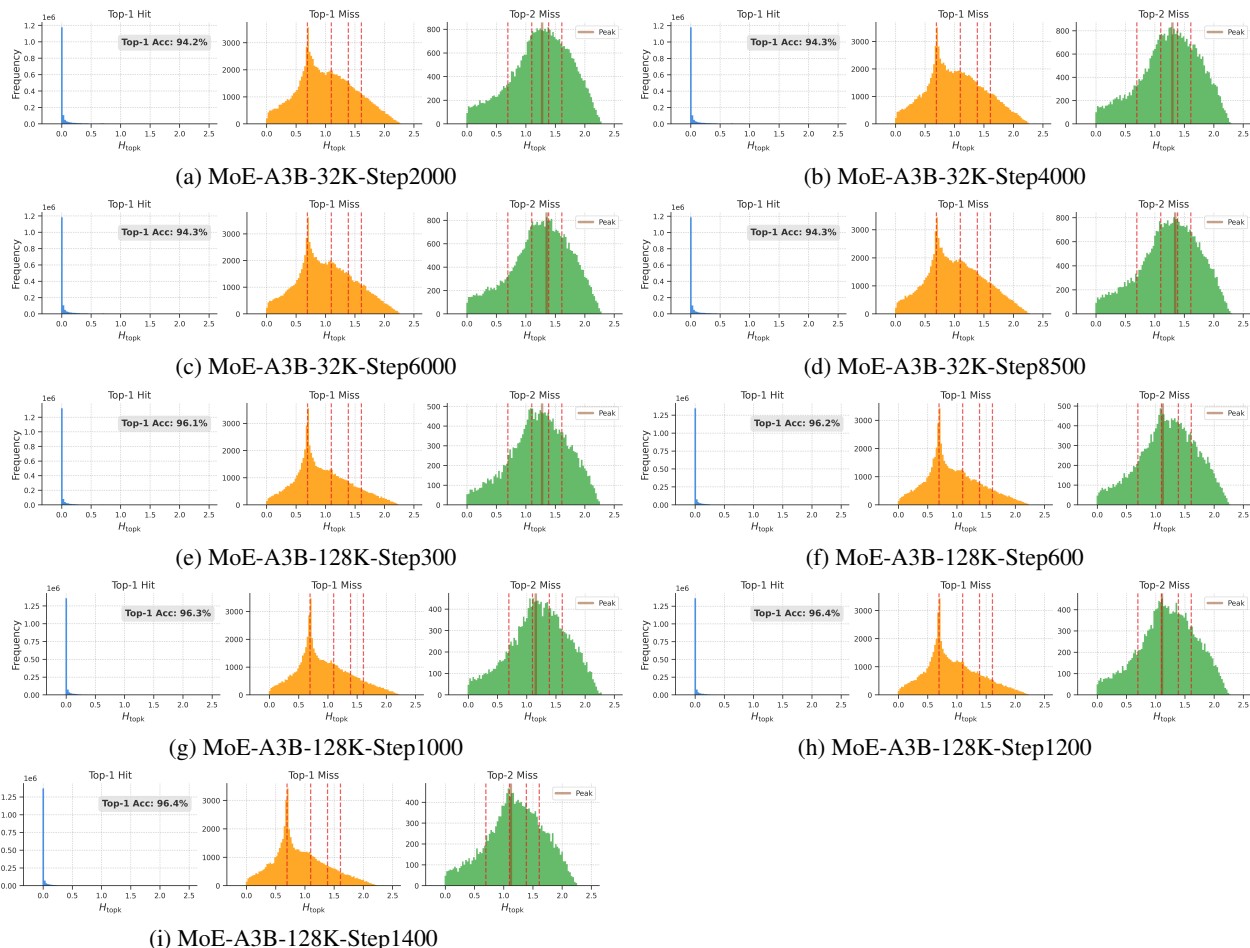

*Figure 11.* Comprehensive evolution of token entropy distributions for the MoE-A3B model. This figure visualizes the entropy landscape across a dense sequence of checkpoints, covering the 32K mid-training phase and the subsequent 128K context extension phase. As in the main text, red dashed lines indicate reference entropy levels corresponding to $\ln 2, \ln 3, \ln 4$, and $\ln 5$.

## E. Generalizability of the Entropy Shift Phenomenon

To address potential concerns that the "Shift to $\ln 3$" phenomenon might be an artifact specific to Mixture-of-Experts (MoE) architectures or exclusive to software engineering tasks, we conducted extended analyses across different model architectures and reasoning domains.

### E.1. Architectural Generalizability.

We evaluated the entropy distributions of Qwen2.5-72B-Base (a Dense architecture) and DeepSeek-V3-Base (a MoE architecture) on the SWE-BENCH trajectories. As shown in Figure 13, both architectures exhibit distinct multi-modal

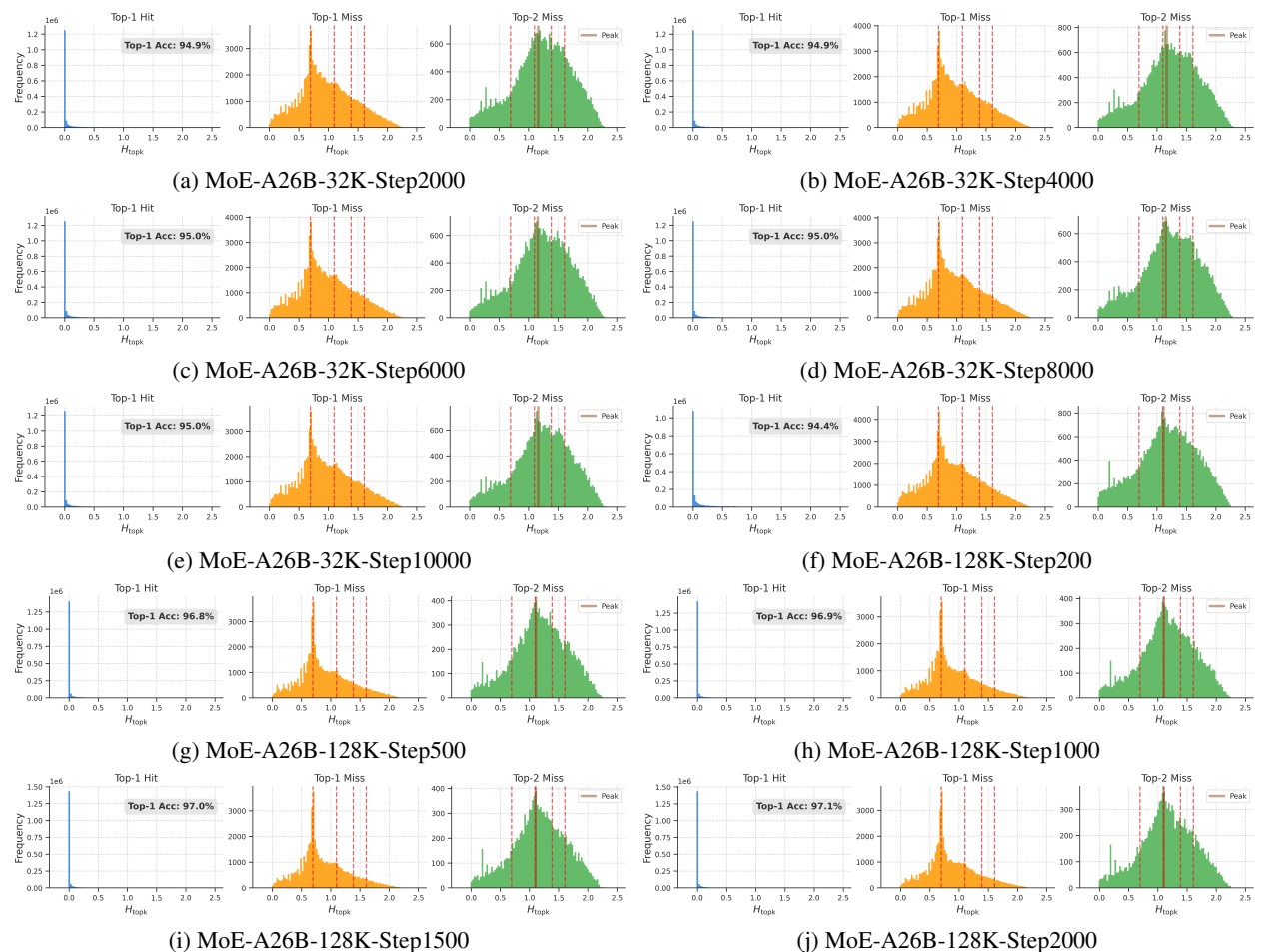

*Figure 12.* Comprehensive evolution of token entropy distributions for the MoE-A26B model. This figure extends the analysis in the main text by visualizing the entropy landscape across a dense sequence of checkpoints during both the 32K mid-training phase and the subsequent 128K context extension phase. As in the main text, red dashed lines indicate reference entropy levels corresponding to $\ln 2, \ln 3, \ln 4,$ and $\ln 5$.

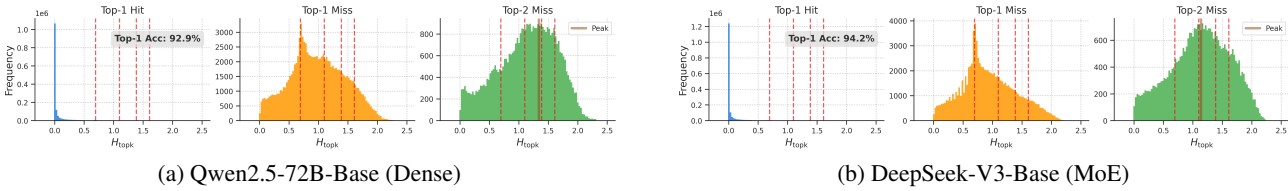

*Figure 13.* Top-10 Entropy distribution of Qwen2.5-72B-Base (Dense) and DeepSeek-V3-Base (MoE) on curated Action tokens. Both architectures consistently exhibit multi-modal peaks at $\ln 1, \ln 2,$ and $\ln 3,$ and demonstrate a "Shift to $\ln 3$" trend. This confirms that the observed entropy distribution is a robust reflection of reasoning behavior, rather than an artifact specific to MoE architectures.

entropy peaks at $\ln 1, \ln 2,$ and $\ln 3.$ Notably, DeepSeek-V3 shows a more pronounced shift towards $\ln 3$ compared to Qwen2.5. This confirms that the phenomenon reflects widespread reasoning behavior across architectures, rather than being an MoE artifact.

### E.2. Domain Generalizability (Mathematical Reasoning).

We further extended our analysis to natural language and mathematical reasoning by evaluating 5,000 rigorous Math QA samples. We analyzed the entropy distributions of the answer tokens across four base models: Qwen2.5-72B-Base, DeepSeek-V3-Base, MoE-A3B-128K-Base, and MoE-A26B-128K-Base. Across all four models, we consistently observed

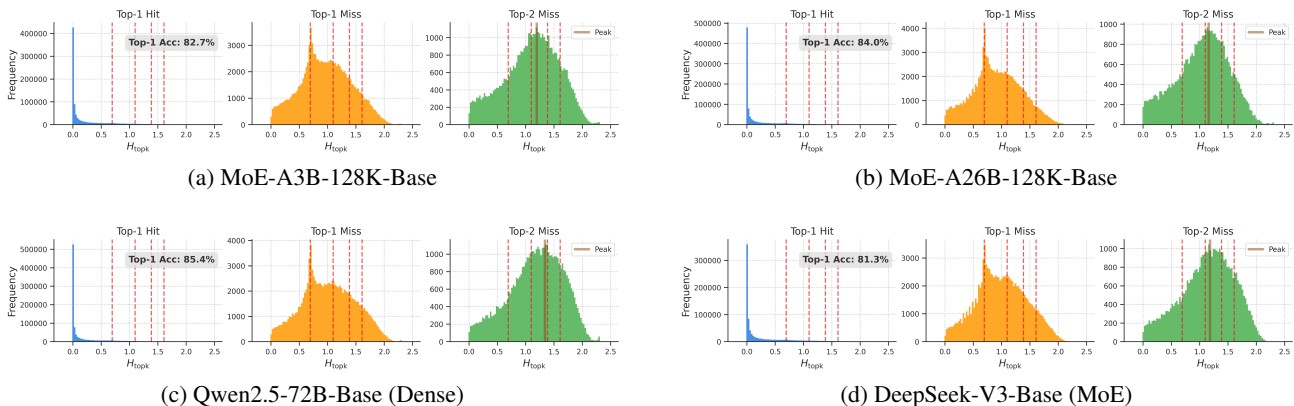

(a) MoE-A3B-128K-Base

(b) MoE-A26B-128K-Base

(c) Qwen2.5-72B-Base (Dense)

(d) DeepSeek-V3-Base (MoE)

*Figure 14.* Top-10 entropy distributions of MoE-A3B-128K-Base, MoE-A26B-128K-Base, Qwen2.5-72B-Base (Dense), and DeepSeek-V3-Base (MoE) calculated on the answer tokens of 5,000 rigorous Math QA samples. Across all four models, we observe consistent and prominent multi-modal peaks at $\ln 1$, $\ln 2$, and $\ln 3$, along with the "Shift to $\ln 3$" trend. This demonstrates that the entropy peaking phenomenon generalizes well beyond coding tasks to mathematical reasoning domains.

clear multi-modal peaks at $\ln 1$, $\ln 2$, and $\ln 3$, along with the "Shift to $\ln 3$" trend.

Based on our empirical observations, we hypothesize that the $\ln 2$ peak represents general reasoning (commonly found in natural language texts lacking strict structural logic), while the $\ln 3$ peak signifies rigorous logical reasoning (prominent in code or highly structured mathematical derivations). These results, illustrated in Figure 14, confirm that our entropy-based threshold methodology can be robustly extended to other reasoning domains.

## F. Peak Identification via Kernel Density Estimation

To accurately capture the dominant trend in the entropy distribution of Non-Top-2 tokens, we identify the global mode (peak) using Kernel Density Estimation (KDE). Unlike simple histogram binning, which is sensitive to bin width and origin, KDE provides a smooth, continuous estimate of the underlying probability density function (PDF).

Let $\mathcal{D} = \{H_1, H_2, \ldots, H_n\}$ denote the set of observed entropy values for the Non-Top-2 tokens, where $n$ is the sample size. The kernel density estimator $\hat{f}_h(H)$ at any entropy value $H$ is defined as:

$$\hat{f}_h(H) = \frac{1}{nh} \sum_{i=1}^{n} K\left(\frac{H - H_i}{h}\right) \tag{18}$$

where:

- $K(\cdot)$ is the **kernel function**, which determines the shape of the weight distribution around each data point.

- $h > 0$ is the **bandwidth** smoothing parameter, controlling the trade-off between bias and variance.

In our implementation, we employ a **Gaussian kernel**:

$$K(u) = \frac{1}{\sqrt{2\pi}} e^{-\frac{1}{2}u^2} \tag{19}$$

Substituting the Gaussian kernel into the estimator yields:

$$\hat{f}_h(H) = \frac{1}{nh\sqrt{2\pi}} \sum_{i=1}^{n} \exp\left(-\frac{(H - H_i)^2}{2h^2}\right) \tag{20}$$

We determine the bandwidth $h$ automatically (e.g., using Scott's Rule) to ensure optimal smoothing. Finally, the peak of the distribution, $H_{\text{peak}}$, is identified by solving the optimization problem over the domain of observed entropies:

$$H_{\text{peak}} = \underset{H \in [\min(\mathcal{D}), \max(\mathcal{D})]}{\arg\max} \hat{f}_h(H) \tag{21}$$

Numerically, we evaluate $\hat{f}_h(H)$ on a dense grid spanning the range of the data to locate the global maximum.

## G. Case Studies: Specific Entropy Instances

In this section, we provide concrete examples corresponding to the distinct entropy peaks discussed in the main text. Specifically, we analyze the MoE-A26B model to illustrate the nature of these high-entropy states. For the $\ln 2$ and $\ln 3$ peaks, we select instances from the filtered *Action* tokens, reflecting logical decision points in SWE tasks. Conversely, for the $\ln 10$ peak, we draw examples from the *Observation* tokens to demonstrate the aleatoric uncertainty inherent in environmental feedback. We present five representative instances for each peak type. For each instance, we detail the target token, its local context, the calculated entropy, and the probability distribution of the Top-10 candidates. Note that the probabilities reported here are the raw model outputs and have not been re-normalized within the Top-10 set.

### G.1. Instances of $\ln 2 \approx 0.6931$ Peak

**Instance 1  Target Token:** `test`      **Entropy:** $0.6932$

**Context Window (Truncated)**

```
1        all_passed = True
2        for test_name, passed in results:
3            status = "PASS" if passed else "FAIL"
4            print(f"{test_name}: {status}")
5            if not passed:
6                all_passed = False
```

*Table 2.* Top-10 Prediction Distribution for Instance 1 ($\ln 2$ Peak).

| Rank | Token | Prob. | Rank | Token | Prob. |
|------|-------|-------|------|-------|-------|
| 1 | status | 0.5000 | 6 | pad | 0.0000 |
| 2 | test | 0.5000 | 7 | status | 0.0000 |
| 3 | name | 0.0000 | 8 | passed | 0.0000 |
| 4 | ' | 0.0000 | 9 | stat | 0.0000 |
| 5 | ': | 0.0000 | 10 | t | 0.0000 |

**Instance 2  Target Token:**  `'__`      **Entropy:** $0.6931$

**Context Window (Truncated)**

```
1  if __name__ == '__main__':
2      print("=" * 60)
3      print("REPRODUCING AUTODISCOVER_MODULES ISSUE")
```

*Table 3.* Top-10 Prediction Distribution for Instance 2 ($\ln 2$ Peak).

| Rank | Token | Prob. | Rank | Token | Prob. |
|------|-------|-------|------|-------|-------|
| 1 | "__ | 0.5000 | 6 | '_ | 0.0000 |
| 2 | '__ | 0.5000 | 7 | ' | 0.0000 |
| 3 | ' | 0.0000 | 8 | "_ | 0.0000 |
| 4 | __ | 0.0000 | 9 |  | 0.0000 |
| 5 | " | 0.0000 | 10 | main | 0.0000 |

**Instance 3   Target Token:** 1      **Entropy:** 0.6936

**Context Window (Truncated)**

```
1 <function=execute_bash>
2   <parameter=command>find /testbed -name "settings.py" | head -1 0</parameter>
3 </function>
```

*Table 4.* Top-10 Prediction Distribution for Instance 3 (ln 2 Peak).

| Rank | Token | Prob. | Rank | Token | Prob. |
|------|-------|-------|------|-------|-------|
| 1 | 5 | 0.5000 | 6 | 0 | 0.0000 |
| 2 | 1 | 0.5000 | 7 | 4 | 0.0000 |
| 3 | 3 | 0.0000 | 8 | 7 | 0.0000 |
| 4 | 2 | 0.0000 | 9 | 6 | 0.0000 |
| 5 | 8 | 0.0000 | 10 | 9 | 0.0000 |

**Instance 4   Target Token:**   the      **Entropy:** 0.6936

**Context Window (Truncated)**

```
1      # Check if the message includes database alias information
2      if 'default' in str(e):
3          print("Database alias 'default' is included in the error message")
4      else:
5          print("Database alias 'default' is missing from the error message")
```

*Table 5.* Top-10 Prediction Distribution for Instance 4 (ln 2 Peak).

| Rank | Token | Prob. | Rank | Token | Prob. |
|------|-------|-------|------|-------|-------|
| 1 | error | 0.5000 | 6 | errorMessage | 0.0000 |
| 2 | the | 0.5000 | 7 | ERROR | 0.0000 |
| 3 | message | 0.0000 | 8 | exception | 0.0000 |
| 4 | current | 0.0000 | 9 | ( | 0.0000 |
| 5 |  | 0.0000 | 10 | Error | 0.0000 |

**Instance 5   Target Token:** 1      **Entropy:** 0.6936

**Context Window (Truncated)**

```
1 <function=str_replace_editor>
2   <parameter=command>view</parameter>
3   <parameter=path>/testbed/django/utils/termcolors.py</parameter>
4   <parameter=view_range>[1 94, 219]</parameter>
5 </function>
```

*Table 6.* Top-10 Prediction Distribution for Instance 5 (ln 2 Peak).

| Rank | Token | Prob. | Rank | Token | Prob. |
|------|-------|-------|------|-------|-------|
| 1 | 2 | 0.5000 | 6 | 4 | 0.0000 |
| 2 | 1 | 0.5000 | 7 | 5 | 0.0000 |
| 3 | 8 | 0.0000 | 8 | 7 | 0.0000 |
| 4 | 3 | 0.0000 | 9 | 6 | 0.0000 |
| 5 | 9 | 0.0000 | 10 | 0 | 0.0000 |

## G.2. Instances of $\ln 3 \approx 1.0986$ Peak

**Instance 1**  **Target Token:** `if`  **Entropy:** 1.0988

**Context Window (Truncated)**

```
1  def test_trailing_slash_validation_without_common_middleware(self):
2      """
3      Test that flatpage form does not require trailing slash when
4      CommonMiddleware is not present, even if APPEND_SLASH=True.
5      """
6      form_data = {
7          'url': '/no_trailing_slash',
```

*Table 7.* Top-10 Prediction Distribution for Instance 1 (ln 3 Peak).

| Rank | Token | Prob. | Rank | Token | Prob. |
|------|-------|-------|------|-------|-------|
| 1 | with | 0.3333 | 6 | in | 0.0000 |
| 2 | when | 0.3333 | 7 | without | 0.0000 |
| 3 | if | 0.3333 | 8 | for | 0.0000 |
| 4 | APP | 0.0000 | 9 | WITH | 0.0000 |
| 5 | though | 0.0000 | 10 | on | 0.0000 |

**Instance 2**  **Target Token:** `assert`  **Entropy:** 1.1025

**Context Window (Truncated)**

```
1      backwards = graph.backwards_plan(('app_a', '0001'))
2
3      assert forwards == [('app_a', '0001')], f"Expected [('app_a', '0001')], got {
   forwards}"
4      assert backwards == [('app_a', '0001')], f"Expected [('app_a', '0001')], got {
   backwards}"
```

*Table 8.* Top-10 Prediction Distribution for Instance 2 (ln 3 Peak).

| Rank | Token | Prob. | Rank | Token | Prob. |
|------|-------|-------|------|-------|-------|
| 1 | print | 0.3332 | 6 | result | 0.0000 |
| 2 | expected | 0.3332 | 7 | expect | 0.0000 |
| 3 | assert | 0.3332 | 8 | forward | 0.0000 |
| 4 | # | 0.0003 | 9 | success | 0.0000 |
| 5 | forwards | 0.0000 | 10 | for | 0.0000 |

**Instance 3**    **Target Token:** 3      **Entropy:** 1.1027

**Context Window (Truncated)**

```
1      # Test cases with various inputs
2      test_cases = [
3          # Positive timedeltas
4          (datetime.timedelta(hours=0), "Zero offset"),
5          (datetime.timedelta(hours=1), "1 hour positive"),
6          (datetime.timedelta(hours=12), "12 hours positive"),
7          (datetime.timedelta(minutes=30), "30 minutes positive"),
8          (datetime.timedelta(hours=5, minutes= 3 0), "5.5 hours positive"),
9
10         # Negative timedeltas
11         (datetime.timedelta(hours=-1), "1 hour negative"),
```

*Table 9.* Top-10 Prediction Distribution for Instance 3 (ln 3 Peak).

| Rank | Token | Prob. | Rank | Token | Prob. |
|------|-------|-------|------|-------|-------|
| 1 | 1 | 0.3332 | 6 | 2 | 0.0001 |
| 2 | 4 | 0.3332 | 7 | 0 | 0.0001 |
| 3 | 3 | 0.3332 | 8 | 8 | 0.0000 |
| 4 | 9 | 0.0001 | 9 | 7 | 0.0000 |
| 5 | 5 | 0.0001 | 10 | 6 | 0.0000 |

**Instance 4**    **Target Token:** 3      **Entropy:** 1.1047

**Context Window (Truncated)**

```
1  <function=execute_bash>
2    <parameter=command>grep -n -C 3 "body.*=" /testbed/tests/mail/tests.py | head -20</
       parameter>
3  </function>
```

*Table 10.* Top-10 Prediction Distribution for Instance 4 (ln 3 Peak).

| Rank | Token | Prob. | Rank | Token | Prob. |
|------|-------|-------|------|-------|-------|
| 1 | 1 | 0.3331 | 6 | 0 | 0.0001 |
| 2 | 5 | 0.3331 | 7 | 6 | 0.0001 |
| 3 | 3 | 0.3331 | 8 | 8 | 0.0001 |
| 4 | 2 | 0.0001 | 9 | 7 | 0.0001 |
| 5 | 9 | 0.0001 | 10 | 4 | 0.0000 |

**Instance 5**    **Target Token:** 0      **Entropy:** 1.0998

**Context Window (Truncated)**

```
1  <function=execute_bash>
2    <parameter=command>cd /testbed && python -m django test tests.migrations.
       test_autodetector --settings=tests.test_sqlite -v 0 </parameter>
3  </function>
```

*Table 11.* Top-10 Prediction Distribution for Instance 5 (ln 3 Peak).

| Rank | Token | Prob. | Rank | Token | Prob. |
|------|-------|-------|------|-------|-------|
| 1 | 2 | 0.3333 | 6 | 6 | 0.0000 |
| 2 | 1 | 0.3333 | 7 | 8 | 0.0000 |
| 3 | 0 | 0.3333 | 8 | 9 | 0.0000 |
| 4 | 3 | 0.0001 | 9 | 4 | 0.0000 |
| 5 | 7 | 0.0000 | 10 | 5 | 0.0000 |

## G.3. Instances of $\ln 10 \approx 2.3025$ Peak (Observation Tokens)

**Instance 1**   **Target Token:** 6     **Entropy:** 2.3026

**Context Window (Truncated)**

```
1 Traceback (most recent call last):
2   File "/testbed/./tests/runtests.py", line 76 6 , in <module>
3     failures = django_tests(
4   File "/testbed/./tests/runtests.py", line 425, in django_tests
5     failures = test_runner.run_tests(test_labels)
```

*Table 12.* Top-10 Prediction Distribution for Instance 1 (ln 10 Peak).

| Rank | Token | Prob. | Rank | Token | Prob. |
|------|-------|-------|------|-------|-------|
| 1 | 7 | 0.1000 | 6 | 0 | 0.1000 |
| 2 | 6 | 0.1000 | 7 | 2 | 0.1000 |
| 3 | 4 | 0.1000 | 8 | 3 | 0.1000 |
| 4 | 5 | 0.1000 | 9 | 8 | 0.1000 |
| 5 | 1 | 0.1000 | 10 | 9 | 0.1000 |

**Instance 2**   **Target Token:** 6     **Entropy:** 2.3026

**Context Window (Truncated)**

```
1 Subject: Test Subject
2 From: fromexample.comTo:  toexample.com
3 Date: Wed, 13 Aug 2025 10:40:57 -0000
4 Message-ID: <175508165790.188.2743415 6 69237999724
    b0203f921b37>--===============1272743200782223159==
```

*Table 13.* Top-10 Prediction Distribution for Instance 2 (ln 10 Peak).

| Rank | Token | Prob. | Rank | Token | Prob. |
|------|-------|-------|------|-------|-------|
| 1 | 7 | 0.1000 | 6 | 0 | 0.1000 |
| 2 | 6 | 0.1000 | 7 | 2 | 0.1000 |
| 3 | 4 | 0.1000 | 8 | 3 | 0.1000 |
| 4 | 5 | 0.1000 | 9 | 8 | 0.1000 |
| 5 | 1 | 0.1000 | 10 | 9 | 0.1000 |

**Instance 3**   **Target Token:** 6     **Entropy:** 2.3026

**Context Window (Truncated)**

```
1 ERROR tests/auth_tests/test_models.py - django.core.exceptions.ImproperlyConf...
2 !!!!!!!!!!!!!!!!!!!! Interrupted: 1 error during collection !!!!!!!!!!!!!!!!!!!!
3 ============================== 1 error in 0.36 s ==============================
4
5 [STDERRExit code: Error: Exit code 1
```

*Table 14.* Top-10 Prediction Distribution for Instance 3 (ln 10 Peak).

| Rank | Token | Prob. | Rank | Token | Prob. |
|------|-------|-------|------|-------|-------|
| 1 | 7 | 0.1000 | 6 | 0 | 0.1000 |
| 2 | 6 | 0.1000 | 7 | 2 | 0.1000 |
| 3 | 4 | 0.1000 | 8 | 3 | 0.1000 |
| 4 | 5 | 0.1000 | 9 | 8 | 0.1000 |
| 5 | 1 | 0.1000 | 10 | 9 | 0.1000 |

**Instance 4   Target Token:** 6      **Entropy:** 2.3026

**Context Window (Truncated)**

```
1 -rw-r--r-- 1 root root  832 Aug 12 16:26 /testbed/tests/template_tests/filter_tests/
    test_json_script.py
2 -rw-r--r-- 1 root root 2288 Aug 12 16:26 /testbed/tests/template_tests/filter_tests/
    test_pluralize.py
3 -rw-r--r-- 1 root root 1 654 Aug 12 16:26 /testbed/tests/template_tests/filter_tests/
    test_slice.py
4 -rw-r--r-- 1 root root  861 Aug 12 16:26 /testbed/tests/template_tests/filter_tests/
    test_truncatechars.py
```

*Table 15.* Top-10 Prediction Distribution for Instance 4 (ln 10 Peak).

| Rank | Token | Prob. | Rank | Token | Prob. |
|------|-------|-------|------|-------|-------|
| 1 | 7 | 0.1000 | 6 | 0 | 0.1000 |
| 2 | 6 | 0.1000 | 7 | 2 | 0.1000 |
| 3 | 4 | 0.1000 | 8 | 3 | 0.1000 |
| 4 | 5 | 0.1000 | 9 | 8 | 0.1000 |
| 5 | 1 | 0.1000 | 10 | 9 | 0.1000 |

**Instance 5   Target Token:** 6      **Entropy:** 2.3026

**Context Window (Truncated)**

```
1 466-    issues.
2 467-    """
3 468-
4 469-    algorithm = "bcrypt_sha256"
```

*Table 16.* Top-10 Prediction Distribution for Instance 5 (ln 10 Peak).

| Rank | Token | Prob. | Rank | Token | Prob. |
|------|-------|-------|------|-------|-------|
| 1 | 7 | 0.1000 | 6 | 0 | 0.1000 |
| 2 | 6 | 0.1000 | 7 | 2 | 0.1000 |
| 3 | 4 | 0.1000 | 8 | 3 | 0.1000 |
| 4 | 5 | 0.1000 | 9 | 8 | 0.1000 |
| 5 | 1 | 0.1000 | 10 | 9 | 0.1000 |

