# OpenReview forum: "HE-SNR: Uncovering Latent Logic via Entropy for Guiding Mid-Training on SWE-bench"
_ICML.cc/2026/Conference — ICML 2026 regular_

### Official Review · Reviewer_mYxv · 2026-03-06

**Soundness:** 2
**Presentation:** 2
**Significance:** 2
**Originality:** 3
**Overall Recommendation:** 4
**Confidence:** 4

**Summary:**

This paper focuses on LLM evaluation and introduces HE SNR, a mid training metric that leverages the Entropy Compression Hypothesis to predict downstream SWE bench capabilities while mitigating the context extension tax.

**Compliance With Llm Reviewing Policy:**

Affirmed.

**Final Justification:**

The authors have addressed my core concerns by providing the requested token filtering ablation study and validating the metric's threshold.

**Key Questions For Authors:**

- Could you provide ablation studies showing the isolated impact of the AST based comment removal versus the formatting noise reduction on the final metric correlation?
- Is there a theoretical justification for why the model compresses uncertainty to three candidates rather than other low order configurations in these coding tasks?
- *Just for discussion*: OpenAI recently detailed why they no longer evaluate on SWE-bench Verified (see https://openai.com/zh-Hans-CN/index/why-we-no-longer-evaluate-swe-bench-verified/), and how do you view the long term relevance and adaptability of benchmarks/metrics in this community?

**Limitations:**

yes

**Strengths And Weaknesses:**

## Strengths
- The paper discusses entropy compression, providing a theoretical lens to differentiate rote memorization from deliberative reasoning.
- The proposed HE SNR metric demonstrates robust empirical correlation with downstream performance, overcoming the long context penalty inherent in standard perplexity.
- The data filtering pipeline isolates functional Action tokens, offering a pragmatic approach to evaluate latent capabilities without stylistic interference.

## Weaknesses
- My main concerns lie in the generalizability of the static entropy threshold and the strong reliance on a specialized data curation pipeline. The fixed boundary is derived from a narrow set of Python and XML artifacts within SWE bench, raising doubts about whether this metric can scale to domains with different inherent uncertainty structures or broader mathematical reasoning tasks without manual recalibration.
- The analysis of the Alignment Tax attributes high entropy degradation to stylistic regularization, overlooking the possibility that suboptimal SFT data quality or hyperparameter tuning caused the performance drop.
- The evaluation is limited to a specific family of Mixture of Experts models, leaving it unclear if the observed *Shift to ln 3* phenomenon applies to dense architectures.

---

> ### Author Rebuttal · Authors · 2026-03-30
>
> We sincerely thank the reviewer for the rigorous and insightful feedback. We have carefully addressed your concerns with new experiments and analyses.
>
> ---
>
> **1. Response to W1 & Q1: Ablation Study on Token Filtering**
>
> To reliably predict post-SFT performance, our metric isolates SFT-invariant signals by filtering stylistic artifacts. As shown in Table 1, shifting from "Thinking" to "Action" tokens drastically improves ranking correlation (*τ*: 0.5192 → 0.9440), peaking at 0.9794 with full filtering. This is because "Thinking" tokens are heavily dictated by SFT templates and correlate weakly with ultimate task success.
>
> **Table 1: Ablation Study on Token Filtering Strategies.** Pearson *r* measures linear fit; Kendall *τ* captures ranking consistency. Our full pipeline achieves the optimal performance across both metrics.
>
> | Token Type | Filtering Strategy | Pearson *r* | Kendall *τ* |
> |:---|:---|:---:|:---:|
> | Thinking | None (Raw Tokens) | 0.5581 | 0.5192 |
> | Action | None (Raw Action Tokens) | 0.9666 | 0.9440 |
> | Action | + Remove XML formatting tags | 0.9526 | 0.9558 |
> | Action | + Remove redundant whitespace & symbols | 0.9520 | 0.9676 |
> | Action | + AST-based comment removal | 0.9649 | 0.9794 |
>
> ---
>
> **2. Response to W1 (Continued): Generalizability to Math Reasoning & Threshold Robustness**
>
> We conducted a new experiment using 5,000 rigorous Math QA samples across four base models (Qwen2.5-72B, DeepSeek-V3, MoE-S-128K, MoE-L-128K). As shown in [Updated Results 2](https://anonymous.4open.science/r/2E50/entropy_math.md), all models exhibit clear multi-modal peaks at ln 1, ln 2, and ln 3, along with the "Shift to ln 3" trend.
>
> The sensitivity analysis in [Updated Results 3](https://anonymous.4open.science/r/2E50/correlation_vs_threshold.md) shows our observation-based threshold (ε = (ln 3 + ln 4) / 2) is remarkably close to the optimal peak for both metrics. The correlation curves show a robust plateau around this value, proving our metric is not overfitted to a single "magic number"—any value within this neighborhood yields highly competitive correlations. This confirms that our entropy-based threshold methodology can be robustly extended to other reasoning domains. We have added this discussion to the *Limitations* section.
>
> ---
>
> **3. Response to W2: Scope of the Alignment Tax**
>
> We sincerely thank the reviewer for pointing out this issue. We acknowledge that our claims were overly strong and required further experimental validation. We have revised our statements throughout the paper: rather than claiming a "universal" intrinsic source, we now explicitly state that our hypothesis *"offers a novel perspective on the alignment tax specifically observed during SFT for complex agentic tasks."*
>
> Regarding suboptimal SFT as an alternative explanation: our evaluated models achieve a highly competitive resolve rate of up to 68% on SWE-bench, which necessitates rigorous data curation. Therefore, it is highly improbable that the observed entropy degradation is merely an artifact of poor tuning.
>
> ---
>
> **4. Response to W3: The "Shift to ln 3" & Architectural Generalizability**
>
> We extended our analysis to **Qwen2.5-72B-Base (Dense)** and **DeepSeek-V3-Base (MoE)**. As shown in [Updated Results 1](https://anonymous.4open.science/r/2E50/Generalizability.md), both architectures exhibit multi-modal entropy peaks at ln 1, ln 2, and ln 3. DeepSeek-V3 shows a more pronounced shift than Qwen2.5. This confirms the phenomenon reflects widespread reasoning behavior across architectures, not an MoE artifact.
>
> ---
>
> **5. Response to Q2: Theoretical Justification for ln 3**
>
> We thank the reviewer for this insightful question. In Appendix C, we show that the maximum entropy of a *k*-candidate uniform distribution is ln *k*, providing the mathematical basis for distinct peaks at these values. The ln 3 peak consistently emerges in both code and math, suggesting that at critical decision points, a model balances among three competing logical paths, naturally converging to three candidates. In contrast, ln 2 dominates in less structured natural language. We acknowledge this remains a hypothesis, and whether higher-order peaks emerge in more complex tasks is left for future work.
>
> ---
>
> **6. Response to Q3: Discussion on OpenAI's Critique and the Future of Metrics**
>
> Despite OpenAI's critiques of SWE-bench Verified regarding brittle execution-based grading and data contamination, HE-SNR retains its fundamental diagnostic value:
>
> - **From Execution to Internal Representation:** HE-SNR bypasses fragile sandbox environments by extracting signals from internal token probability distributions, completely decoupling "true reasoning capability" from "test-script compatibility."
>
> - **Longevity of Diagnostic Methodology:** As benchmarks retire due to contamination, HE-SNR's entropy-based framework remains adaptable. Even as the community shifts to SWE-bench Pro or future benchmarks, our diagnostic methodology maintains its core value.

---

> > ### Author Rebuttal · Reviewer_mYxv · 2026-04-03
> >
> > The authors have addressed my core concerns by providing the requested token filtering ablation study and validating the metric's threshold. I am willing to increase my score.

---

> > > ### Author Response · Authors · 2026-04-04
> > >
> > > Dear Reviewer,
> > >
> > > Thank you very much for your time and for acknowledging our rebuttal. We are thrilled to hear that the token filtering ablation study and the metric threshold validation have fully addressed your core concerns.
> > >
> > > We also deeply appreciate your willingness to increase the score. We noticed that the numerical score in the system currently remains unchanged. We are sending this gentle note just in case updating the score requires a separate submission step in the system, or if there is a system delay.
> > >
> > > Thank you again for your constructive feedback, which has greatly helped us improve our work!
> > >
> > > Best regards,
> > >
> > > Authors

---

### Official Review · Reviewer_x84C · 2026-03-09

**Soundness:** 3
**Presentation:** 3
**Significance:** 4
**Originality:** 3
**Overall Recommendation:** 5
**Confidence:** 4

**Summary:**

This paper addresses the limitations of standard metrics like Perplexity (PPL) in guiding the "mid-training" phase of Large Language Models (LLMs), particularly when extending context windows. It introduces HE-SNR, a metric that targets "high-entropy" decision points where models exhibit "reasonable hesitation" among a small set of candidates. This is grounded in the Entropy Compression Hypothesis, which posits that intelligence is the capacity to refine uncertainty into a few plausible options.

**Compliance With Llm Reviewing Policy:**

Affirmed.

**Final Justification:**

The reviewer clearly addressed my questions. I decide to keep my previous score.

**Key Questions For Authors:**

Does the "Shift to ln3" phenomenon hold across non-coding logical domains, such as symbolic mathematics or formal logic? In fact, you identify a distinct peak at ln3 as a signature of superior reasoning. Is this value of k=3 a fundamental property of the SWE-BENCH tasks, or do you expect this "reasoning boundary" to shift for simpler tasks (e.g., ln2 for binary logic) or more complex ones (e.g., ln4 or ln5 for multi-step mathematical proofs)?

You observe that during context extension, Top-1 accuracy and PPL degrade while Top-10 accuracy remains stable. This suggests the model’s "knowledge" remains within the candidate set but its "calibration" is disrupted. Could the "Long-Context Tax" be mitigated through simple post-hoc calibration techniques (like temperature scaling) rather than requiring a new metric like HE-SNR?

**Limitations:**

yes

**Strengths And Weaknesses:**

Strengths: The "Shift to ln3" observation provides a novel theoretical lens for understanding reasoning growth. The work includes industrial-scale validation on Mixture-of-Experts (MoE) models and demonstrates that HE-SNR is robust against the "Long-Context Tax" that typically degrades PPL.

Weaknesses:
1. The current "Action" token filtering relies on a conservative strategy that may still be influenced by SFT-specific artifacts.

2. The critical entropy threshold is set at the midpoint between ln3 and ln4. While empirically effective for this study, the paper does not provide a theoretical proof that this specific threshold is optimal for non-coding logical domains.

3. While excluding "Thoughts" reduces stylistic noise, it may also discard valuable latent reasoning steps. As "reasoning-heavy" models become more prevalent, ignoring the internal CoT might overlook the very tokens where logical breakthroughs occur.

4. The use of AST parsing to strip comments and whitespace ensures a focus on executable logic, but it may penalize models that produce well-documented code, which is a key requirement in real-world software engineering.

---

> ### Author Rebuttal · Authors · 2026-03-30
>
> We sincerely thank the reviewer for the rigorous evaluation and constructive suggestions.
>
> ---
>
> **1. Response to W1: Action Token Filtering Strategy**
>
> We agree our heuristic-based filtering strategy may leave residual SFT artifacts. We conducted an ablation study (Table 1) shows that shifting from "Thinking" to "Action" tokens drastically improves the ranking correlation (*τ*: 0.5192 → 0.9440), and progressive filtering pushes Kendall *τ* to its peak (0.9794).
>
> The most crucial step of our pipeline is removing "Thinking" tokens. Within the "Action" space, after filtering SFT-specific artifacts, we further remove code comments because they inherently resemble NL. While empirically effective, semantic filtering has limitations. Exploring purely mathematical tools (e.g., entropy-based filtering) for automatic artifact isolation is a key future direction, which we have now discussed in the *Limitations* and *Future Work* sections.
>
> **Table 1: Ablation Study on Token Filtering Strategies.** Impact of filtering steps on HE-SNR and SWE-bench correlation. While Pearson *r* measures the linear fit, Kendall *τ* captures the crucial ranking consistency.
>
> | Token Type | Filtering Strategy | Pearson *r* | Kendall *τ* |
> |:---|:---|:---:|:---:|
> | Thinking | None (Raw Tokens) | 0.5581 | 0.5192 |
> | Action | None (Raw Action Tokens) | 0.9666 | 0.9440 |
> | Action | + Remove XML formatting tags | 0.9526 | 0.9558 |
> | Action | + Remove redundant whitespace & symbols | 0.9520 | 0.9676 |
> | Action | + AST-based comment removal | 0.9649 | 0.9794 |
>
> ---
>
> **2. Response to W2 & Q1: Generalizability to Non-Code Tasks**
>
> In [Updated Results 2](https://anonymous.4open.science/r/2E50/entropy_math.md), we tested using 5,000 rigorous Math QA samples across four base models (Qwen2.5-72B, DeepSeek-V3, MoE-S, MoE-L). All four models still exhibit clear multi-modal peaks at ln 1, ln 2, and ln 3, along with the "Shift to ln 3" trend. These results suggest that our findings are not limited to code but can be generalized to a broader range of reasoning domains.
>
> Based on our empirical observations, we hypothesize the following: the ln 2 peak represents *general reasoning* (commonly found in natural language texts lacking strict structural logic), while the ln 3 signifies *rigorous logical reasoning* (code or structured math). Whether higher-order peaks (ln 4, ln 5) emerge in more complex tasks remains an open question for future work.
>
> ---
>
> **3. Response to W3: Filtering "Thought" Tokens**
>
> As shown in Table 1, filtering "Thinking" tokens drastically improves our metric's correlation, because the "thinking" part is heavily influenced by stylistic prompt templates, making its entropy less predictive of ultimate task success than the strictly executable "Action" part. However, we fully agree with your insightful point: as RL-driven "reasoning-heavy" models emerge, internal CoT will contain critical logical breakthroughs. Adapting our metric to evaluate these dense CoT tokens is a vital direction for future work.
>
> ---
>
> **4. Response to W4: Filtering Comments and Whitespace**
>
> For base models, formatting (e.g., predicting varying numbers of whitespaces) and commenting style are highly variable and easily regularized during SFT, and do not affect task correctness. Because comments resemble natural language (similar to "Thinking") and whitespace variations are noise, filtering them is completely justified. Since SFT readily resolves formatting issues, evaluating a base model's agentic potential should strictly focus on its core logical capabilities, which are the true bottlenecks for complex tasks.
>
> ---
>
> **5. Response to Q2: Temperature Scaling vs. HE-SNR**
>
> We highly appreciate your insightful observation that the model's "knowledge" remains intact while its "calibration" is disrupted. However, post-hoc calibration techniques like temperature scaling cannot mitigate this issue or replace HE-SNR, for two fundamental reasons:
>
> 1. **Ineffectiveness for greedy decoding.** Rigorous tasks (e.g., SWE-bench) typically rely on greedy decoding (*T* = 0). Since temperature scaling is a monotonic transformation of logits, it preserves token rankings and cannot recover a correct token that has fallen below the Top-1 position. Conversely, introducing *T* > 0 to shift distributions often injects stochasticity that leads to syntax errors in structured reasoning.
>
> 2. **PPL's inherent flaws in ranking consistency.** Beyond its susceptibility to the Long-Context Tax, standard PPL suffers from extremely poor ranking consistency with downstream agentic performance, as it is heavily skewed by high-frequency stylistic tokens.
>
> Ultimately, post-hoc calibration techniques are inference-time patches. In contrast, HE-SNR is designed as a robust training-time diagnostic metric. We need a reliable "ruler" to accurately evaluate and rank a base model's true logical capabilities during mid-training.

---

> > ### Author Rebuttal · Reviewer_x84C · 2026-04-02
> >
> > The reviewer clearly addressed my questions. I decide to keep my previous score.

---

> > > ### Author Response · Authors · 2026-04-05
> > >
> > > Dear Reviewer,
> > >
> > > Thank you for reviewing our rebuttal and for confirming that our response clearly addressed your questions.
> > >
> > > We are especially grateful for your strong initial evaluation and your positive recognition of our work. Your appreciation of our contributions and your support throughout the review process are highly encouraging to our team.
> > >
> > > Thank you once again for your time and valuable feedback!
> > >
> > > Best regards,
> > >
> > > Authors

---

### Official Review · Reviewer_huSP · 2026-03-13

**Soundness:** 3
**Presentation:** 2
**Significance:** 3
**Originality:** 2
**Overall Recommendation:** 4
**Confidence:** 3

**Summary:**

This paper investigates the robustness of standard evaluation metrics such as perplexity as indicators of downstream task performance for LLM mid-training on SWE agentic tasks (SWE-bench, in particular). The authors formulate the Entropy Compression Hypothesis, which builds upon prior hypotheses that focus on information compression, and then propose HE-SNR, an evaluation metric that aims at better forecasting LLM performance on downstream tasks during training, as well as a data-efficient evaluation protocol. The authors also argue that the proposed hypothesis explains the existence of the alignment tax. Experiments with two proprietary LLMs on SWE-bench show that the proposed metric correlates with downstream model performance on the task.

**Compliance With Llm Reviewing Policy:**

Affirmed.

**Final Justification:**

Updated scores due to rebuttal.

**Key Questions For Authors:**

Please see weakness, and would be helpful to clarify the relationship between the proposed method to other prior works like LongPPL.

**Limitations:**

Yes

**Strengths And Weaknesses:**

Strength:
- Significance: The paper studies an important and timely topic, and the insights would potentially benefit a variety of future work on fine-tuning LLMs for agentic tasks. While the current study focuses on SWE-bench, the insights seems to be, in principle, generalizable to a variety of agentic tasks, which is worth future discussion and research.

- Soundness: The proposed HE-SNR metric is well-motivated and empirically achieves good correlation with downstream performance.

Weakness:
- Originality: There are prior works that invoke a similar observation (e.g., LongPPL [1] and its peer works when eligible), which follow a similar insight that PPL overlooks key tokens and is particularly poorly geared towards long-context scenarios. Without discussion about these prior works and an empirical comparison with eligible baselines, it would be hard for readers to evaluate the originality/significance of the work.

- Presentation: Presentation could be improved; please see “other writing suggestions.” An actionable suggestion would be clearly outlining claims/authors' insights versus universally accepted knowledge, and providing pointers to experiments/citations for both types of statements, respectively.

- Soundness: There are some claimed contributions that need further discussion/experiments. For example, the proposed data filtering mechanism currently is only discussed briefly, and there are limited discussions or comparisons with its potential alternatives. Meanwhile, the authors claim that the work uncovers the intrinsic source of the alignment tax, while the studies are focused only on SWE-bench and SFT, in contrast to the alignment tax observed in other settings such as RLHF and safety alignment.

Other Writing suggestions:

L17, right column: “Therefore, incorporating an SFT phase is mandatory for assessing model performance on SWE-BENCH…”  This is a strong claim against, e.g., RL-based adaptation techniques, and assumes that it is universally accepted that SFT is mandatory. To this end, some citations/discussion would help.

Line 55-63: Gaps in prior work are not entirely clearly articulated. I understand the argument here is that prior metrics are often confined to non-agentic benchmarks, but the authors then raise two implicit points: (1) these non-agentic tools are less dependent on SFT alignment, which itself is not a gap for prior metrics like PPL/BPC; and (2) then the authors pivot to the limitation of existing correlation studies.

Overall, while the readers get a general idea that prior work is limited, the argument here is somewhat scattered, making it hard for readers to establish an evaluation of the credibility of these arguments, backed by either citations or experiments.

Lines 76-78: It would be helpful to have a high-level, but somewhat concrete overview of the authors’ proposal (e.g., “We propose…”), which gives readers more context before highlighting the itemized contributions."

---

> ### Author Rebuttal · Authors · 2026-03-30
>
> We sincerely thank the reviewer for the constructive feedback and the valuable literature recommendations.
>
> ---
>
> **1. Response to W1: The Relationship to LongPPL & Baselines**
>
> We thank the reviewer for highlighting LongPPL and related works. We completely agree that they share the high-level insight that standard PPL overlooks key tokens in long-context scenarios. We have added a dedicated discussion on them in the revised related work section. Crucially, we differentiate our work in two main aspects:
>
> - **Model & Task Focus:** LongPPL studies position bias in instruct models on long-context benchmarks (e.g., RULER). Conversely, HE-SNR analyzes base models via entropy to predict downstream agentic performance.
>
> - **Empirical Comparison:** While LongPPL identifies tokens critical for information retrieval, HE-SNR extracts SFT-invariant signals to predict logical reasoning. Given their fundamentally different evaluation targets (retrieval vs. reasoning), we hypothesize that LongPPL may struggle to predict complex agentic performance. We will add a comparison of the two approaches in the revised manuscript.
>
> ---
>
> **2. Response to W3: Ablation Study on Data Filtering Alternatives**
>
> To establish a reliable metric during the mid-training phase, we must account for the fact that SFT drastically alters the model's output distribution. Our data filtering pipeline is intentionally designed to strip away these SFT-induced artifacts.
>
> We conducted an ablation study comparing our pipeline against alternative token selections. As shown in Table 1, shifting from "Thinking" to "Action" tokens drastically improves the ranking correlation (*τ* increases from 0.5192 to 0.9440). Furthermore, the progressive application of our filtering strategies strictly isolates the "executable logic," pushing Kendall *τ* to its peak (0.9794).
>
> **Table 1: Ablation Study on Token Filtering Strategies.** Impact of filtering steps on HE-SNR and SWE-bench correlation. While Pearson *r* measures the linear fit, Kendall *τ* captures the crucial ranking consistency. Our full pipeline achieves the optimal performance across both metrics.
>
> | Token Type | Filtering Strategy | Pearson *r* | Kendall *τ* |
> |:---|:---|:---:|:---:|
> | Thinking | None (Raw Tokens) | 0.5581 | 0.5192 |
> | Action | None (Raw Action Tokens) | 0.9666 | 0.9440 |
> | Action | + Remove XML formatting tags | 0.9526 | 0.9558 |
> | Action | + Remove redundant whitespace & symbols | 0.9520 | 0.9676 |
> | Action | + AST-based comment removal | 0.9649 | 0.9794 |
>
> ---
>
> **3. Response to W3: Scope of the Alignment Tax**
>
> We completely agree that our current scope is limited to SFT and SWE-bench. We have revised our claims throughout the paper to be more precise: rather than claiming to uncover a "universal" intrinsic source, we now explicitly state that our hypothesis *"offers a novel perspective on the alignment tax specifically observed during SFT for complex agentic tasks."*
>
> ---
>
> **4. Response to W2 and Writing Suggestions**
>
> 1. **L17 (SFT is mandatory):** We have softened the claim to: *"Therefore, incorporating an SFT phase is widely considered a crucial step for assessing..."* and added citations on RL-based alternatives to provide a balanced view.
>
>    Furthermore, applying RL directly to base models ("from-zero RL") is highly challenging for complex, multi-turn tasks like SWE-bench. It is standard practice to perform SFT before RL to establish fundamental instruction-following capabilities. For instance, recent SOTA RL frameworks (SWE-RL [1], SWE-Dev [2]) are built on instruction-tuned models, and Kimi-Dev [3] explicitly applies SFT on base models prior to RL. This strongly supports our premise that an SFT phase is a necessary prerequisite.
>
> 2. **L55-63 (Gaps in prior work):** We sincerely thank you for the structural guidance. Based on your feedback, we have rewritten this paragraph to improve the logical flow, explicitly separating the arguments into two distinct points: (1) why non-agentic metrics fail to capture SFT alignment nuances, and (2) the limitations of existing correlation studies.
>
>    Specifically, we clarify that while prior metrics (e.g., PPL/BPC) are effective for non-agentic benchmarks evaluated directly on base models, our work pioneers metric design for agentic tasks. Because the core challenge here is extracting SFT-invariant signals and no prior agentic metric exists, PPL serves as our only viable baseline.
>
> 3. **L76-78 (High-level overview):** We have added a concrete, high-level overview (*"In this work, we propose..."*) right before the itemized contributions to provide readers with better context.
>
> ---
>
> [1] Wei, Y., et al. "SWE-RL: Advancing LLM reasoning via reinforcement learning on open software evolution." arXiv:2502.18449 (2025).
>
> [2] Wang, H., et al. "SWE-Dev: Building software engineering agents with training and inference scaling." ACL 2025.
>
> [3] Yang, Z., et al. "Kimi-Dev: Agentless training as skill prior for SWE-agents." arXiv:2509.23045 (2025).

---

> > ### Author Rebuttal · Reviewer_huSP · 2026-04-04
> >
> > Appreciate the rebuttal, my concerns are resolved. Scores are updated.

---

> > > ### Author Response · Authors · 2026-04-05
> > >
> > > Dear Reviewer,
> > >
> > > Thank you for taking the time to read our rebuttal and for updating the score. We are delighted to know that our response has fully resolved your concerns.
> > >
> > > We would like to express our special gratitude for your detailed suggestions regarding the writing and the valuable references you recommended. Incorporating these citations and refining the presentation based on your advice has significantly improved the clarity and comprehensiveness of our manuscript.
> > >
> > > Thank you once again for your constructive feedback and support throughout the review process!
> > >
> > > Best regards,
> > >
> > > Authors

---

### Official Review · Reviewer_nDnS · 2026-03-13

**Soundness:** 2
**Presentation:** 3
**Significance:** 3
**Originality:** 3
**Overall Recommendation:** 5
**Confidence:** 2

**Summary:**

This paper proposes HE-SNR, a proxy metric designed to track models' mid-training toward downstream performance on SWE-BENCH, motivated by the claim that standard PPL becomes unreliable under long-context adaptation. Empirically,  the authors show that HE-SNR, computed on a filtered set of high-entropy "action tokens", exhibits a stronger correlation with post-SFT SWE-BENCH performance than conventional PPL-based metrics.  While the empirical finding are promising,  the paper's broader theoretical claims appear to extend beyond what the current evidence fully supports.

**Compliance With Llm Reviewing Policy:**

Affirmed.

**Final Justification:**

I have increased my score.

**Key Questions For Authors:**

See weaknesses.

**Limitations:**

See weaknesses.

**Strengths And Weaknesses:**

Strength:

1. This paper proposed a cheap and reliable mid-training proxy to guide mid-training for SWE-BENCH, which is well motivated and relevant for large-scale training.

2. The empirical study of this paper is clear and compelling. The HE-SNR on filtered action tokens appears noticeably better aligned with downstream SWE-BENCH performance than standard PPL-based alternatives.

Weakness:
1. The claim "shift to $\ln 3$" is not fully convincing as a principle. Although the observed entropy peak is interesting, the current analysis does not rule out the possibility that this specific peak is an artifact of the model family (MoE), the structured data format, or the particular tokenizer used, rather than a reflection of a more general law of reasoning. Also the "Shift to ln3" phenomenon and HE-SNR’s performance are validated primarily on SWE-BENCH, raising questions about generalizability to non-code tasks (e.g., natural language reasoning)  .

2. As most conclusions are drawn from two MoE models and a fixed SWE-BENCH to SFT evaluation pipeline, it remains unclear whether this metric would be suitable to other model architectures and different data distributions. While MoE-L is large (hundreds of B parameters), the study lacks comparison with non-MoE architectures, limiting claims about architectural generalizability .

3. The method proposed in this paper seems relies heavily on hand-crafted choices such as action-only evaluation, aggressive token filtering, top-$10$ truncation. As a result, it is still unclear whether the same metric design would remain effective on other software-engineering benchmarks.

---

> ### Author Rebuttal · Authors · 2026-03-30
>
> We sincerely thank the reviewer for the rigorous and insightful feedback. We have carefully addressed your concerns with new experiments and analyses.
>
> ---
>
> **1. Response to W1 & W2: The "Shift to ln 3" & Architectural Generalizability**
>
> We agree "principle" was too strong and have reframed the "Shift to ln 3" as an empirical observation specific to long-context agentic alignment. To address artifact concerns, we extended our analysis to **Qwen2.5-72B-Base (Dense)** and **DeepSeek-V3-Base (MoE)**. As shown in [Updated Results 1](https://anonymous.4open.science/r/2E50/Generalizability.md), both architectures exhibit multi-modal entropy peaks at ln 1, ln 2, and ln 3. DeepSeek-V3 shows a more pronounced shift than Qwen2.5. This confirms the phenomenon reflects widespread reasoning behavior, rather than an artifact of MoE architectures or specific tokenizers.
>
> Regarding our initial focus on MoE models, we prioritized them because they represent the current state-of-the-art (SOTA) for complex reasoning; nearly all models achieving top-tier performance on SWE-bench utilize 100B+ MoE architectures. Given the massive computational resources required for mid-training/SFT at this scale, our study intentionally prioritized these architectures to ensure the metric is directly applicable to cutting-edge model production.
>
> ---
>
> **2. Response to W3: Robustness of Data Curation & Generalizability**
>
> Regarding "hand-crafted choices", our token filtering and action-only evaluation are systematic necessities to strip away SFT artifacts and isolate executable logic. Table 1 isolates the impact of each filtering step on HE-SNR's correlation with SWE-bench. Shifting from "Thinking" to "Action" tokens drastically improves Kendall *τ* (0.5192 → 0.9440). Progressively applying our filters pushes the correlation to its peak (0.9794).
>
> **Table 1: Ablation Study on Token Filtering.** Impact of filtering steps on HE-SNR and SWE-bench correlation. While Pearson $r$ measures the linear fit, Kendall $\tau$ captures the crucial ranking consistency. Our full pipeline achieves the optimal performance across both metrics.
>
> | Token Type | Filtering Strategy | Pearson *r* | Kendall *τ* |
> |:---|:---|:---:|:---:|
> | Thinking | None (Raw Tokens) | 0.5581 | 0.5192 |
> | Action | None (Raw Action Tokens) | 0.9666 | 0.9440 |
> | Action | + Remove XML tags | 0.9526 | 0.9558 |
> | Action | + Remove whitespace & symbols | 0.9520 | 0.9676 |
> | Action | + AST-based comment removal | 0.9649 | 0.9794 |
>
> Regarding the **Top-10 truncation**, this design mitigates long-tail numerical noise and significantly reduces computational overhead. Since commercial APIs (e.g., OpenAI) restrict log-probability outputs (e.g., top-20), full-vocabulary entropy is impossible. Crucially, because our metric focuses on small *k* values, a Top-10 window is more than sufficient to capture these critical reasoning dynamics. Thus, truncation is an inevitable and practical necessity for real-world evaluation.
>
> Since SWE-bench is the most rigorous SE benchmark and our filtering isolates functional correctness, this metric design should remain robust on other SE benchmarks.
>
> ---
>
> **3. Response to W1 (Continued): Generalizability to Math Reasoning**
>
> We sincerely thank the reviewer for raising this insightful point. To address your concern about natural language and mathematical reasoning, we conducted a new experiment using 5,000 rigorous Math QA samples. We analyzed the entropy distributions of the answer tokens across four base models: Qwen2.5-72B-Base, DeepSeek-V3-Base, MoE-S-128K-Base, and MoE-L-128K-Base.
>
> As shown in [Updated Results 2](https://anonymous.4open.science/r/2E50/entropy_math.md), all four models still exhibit clear multi-modal peaks at ln 1, ln 2, and ln 3, along with the "Shift to ln 3" trend.
>
> Based on our empirical observations, the ln 3 peak consistently emerges in texts requiring rigorous logical reasoning (prominent in code or highly structured mathematical derivations). Furthermore, in [Updated Results 3](https://anonymous.4open.science/r/2E50/correlation_vs_threshold.md), we present a sensitivity analysis showing the impact of different entropy thresholds on the correlation between HE-SNR and downstream SWE-bench scores. As demonstrated, our observation-based threshold (ε = (ln 3 + ln 4) / 2) is remarkably close to the optimal peak for both Pearson and Kendall metrics, confirming the soundness of our metric design.
>
> More importantly, the correlation curves show a robust plateau around this value. Therefore, the methodology of analyzing entropy thresholds is generalizable to non-code tasks, though the specific threshold values may require recalibration based on the domain's inherent uncertainty structure. We have added this discussion to the *Limitations* section.

---

> > ### Author Rebuttal · Reviewer_nDnS · 2026-04-04
> >
> > I thank the authors for the  response and the additional experiments.

---

> > > ### Author Response · Authors · 2026-04-05
> > >
> > > Dear Reviewer,
> > >
> > > Thank you very much for your time and for reviewing our rebuttal. We are very glad to hear that our response and the additional experiments have fully addressed your concerns.
> > >
> > > We deeply appreciate your constructive feedback throughout the review process, which has significantly helped us improve the quality and rigor of our paper.
> > >
> > > Thank you again for your valuable efforts and support!
> > >
> > > Best regards,
> > >
> > > Authors

---

### Decision · Program_Chairs · 2026-04-30

**Decision:**

Accept (regular)

**Comment:**

This paper addresses a practical issue of PPL that it poorly forecasts downstream task performance on complex tasks like SWE-bench. The authors propose HE-SNR metric to resolve this issue that effectively mitigates the "Long-Context Tax" by focusing on high-entropy "hesitation" points. The proposed framework is validated on industrial-scale Mixture-of-Experts (MoE) models across 32K and 128K context windows, and it is demonstrated that HE-SNR has superior robustness and better correlation with post-Supervised Fine-Tuning (SFT) performance.

The proposed methodology is sound and highly practical supported by large-scale numerical validation. During the rebuttal phase, the authors appropriately addressed the question raised by reviewers about the generalizability of the "Shift to $\ln 3$" phenomenon. Furthermore, the authors provide some theoretical discussions to justify the proposal in addition to empirical validations, which makes the submission solid.

For these reasons, this paper provides a valuable contribution to the community and thus I recommend for acceptance.